# Neuronal evidence for good-based economic decisions under variable action costs

Xinying Cai[1,2,3,4] & Camillo Padoa-Schioppa [1,5,6]

Previous work showed that economic decisions can be made independently of spatial contingencies. However, when goods available for choice bear different action costs, the decision necessarily reflects aspects of the action. One possibility is that "stimulus values" are combined with the corresponding action costs in a motor representation, and decisions are then made in actions space. Alternatively, action costs could be integrated with other determinants of value in a non-spatial representation. If so, decisions under variable action costs could take place in goods space. Here, we recorded from orbitofrontal cortex while monkeys chose between different juices offered in variable amounts. We manipulated action costs by varying the saccade amplitude, and we dissociated in time and space offer presentation from action planning. Neurons encoding the binary choice outcome did so well before the presentation of saccade targets, indicating that decisions were made in goods space.

[1] Department of Neuroscience, Washington University in St Louis, St Louis, MO 63110, USA. [2] NYU Shanghai, 1555 Century Avenue, Shanghai 200122, China. [3] Shanghai Key Laboratory of Brain Functional Genomics (Ministry of Education), School of Psychology and Cognitive Science, East China Normal University, Shanghai 200062, China. [4] NYU-ECNU Institute of Brain and Cognitive Science at NYU Shanghai, 3663 Zhongshan Road North, Shanghai 200062, China. [5] Department of Economics, Washington University in St Louis, St Louis, MO 63110, USA. [6] Department of Biomedical Engineering, Washington University in St Louis, St Louis, MO 63110, USA. Correspondence and requests for materials should be addressed to X.C. (email: xinying.cai@nyu.edu)

A ccording to current views, economic choices involve the computation and comparison of subjective values, subjective values integrate multiple determinants relevant to the decision (commodity, quantity, probability, etc.), and this integration takes place in prefrontal regions including the orbitofrontal cortex (OFC)[1–3]. The activity of neurons in the primate OFC is known to be independent of the spatial contingencies of the task, including the action performed by the subject[4,5]. Furthermore, it is widely recognized that decisions between goods take place in this non-spatial representation, whereas other types of decisions are primarily or exclusively action-based[6–8]. However, it remains unclear exactly under what conditions a decision should be conceptualized as "between goods". In other words, it is not clear what decisions take place in a spatial (action-based) versus non-spatial (good-based) representation[9]. A particularly interesting case is that of choices between goods that vary for their action cost. In such conditions, the decision process necessarily takes into account some aspect of the action. Thus, two broad schemes have been put forth. One possibility is that the brain first computes the "stimulus value" (i.e., the subjective value minus the action cost) in a non-spatial representation, and then combines the stimulus value with the corresponding action cost in a spatial representation. In this scheme, decisions under variable action costs would be action-based and take place in premotor regions[6,10]. Alternatively, action costs might be integrated with other determinants of value in a non-spatial representation. In this scheme, decisions under variable action costs could be good-based[3].

A closely related question pertains to the frames of reference in which goods and values are represented. Neurons encoding the subjective value were first observed in the OFC of monkeys choosing between different juices offered in variable amounts. Different groups of cells encoded the value of individual options, the identity of the chosen option and the chosen value[4,11]. In that representation, options were defined by the juice type. In other words, neurons encoding the offer value were associated to a specific juice, and their activity was linearly related to the quantity offered on any given trial. We refer to this reference frame as "commodity-based". Notably, this reference frame was not imposed by the choice task. An equally valid reference frame would have been that in which cells encoding the offer value are associated with a particular location. Subsequent studies suggested that neurons in OFC are flexible, and that the reference frame can adapt to the characteristics of the choice task. For example, in the study of Tsujimoto et al., options were defined uniquely by their spatial location. Some neurons in the OFC encoded the identity of the chosen option in a way that was indistinguishable from a spatial representation[12]. Similar results were also obtained by Abe and Lee[13,14]. More recent data suggest that, under proper circumstances, the reference frame in OFC can be based on a specific trait of the offer such as its informativeness[15] (see Discussion). Taken together, these results suggest that the reference frame in which good identities and values are encoded in OFC may be malleable and adapt to the characteristics of the choice task.

The experiments described here were conducted to assess whether economic decisions under variable action costs can take place in a non-spatial representation (goods space). While designing the choice task, we considered several issues.

First, it is generally difficult to ascertain whether a decision is made in goods space or actions space based on behavior alone. However, this issue may be addressed using neural measures. Specifically, to establish that a decision is good-based, it is necessary to dissociate in time and space the presentation of the offers and the indication of the actions associated with each offer. Previous studies that used this approach focused on decisions under fixed action costs[16,17]. In these studies, subjects were presented two offers at the beginning of each trial; later in the trial, subjects were shown two action targets associated with the two offers. Neuronal activity encoding the choice outcome before presentation of the action targets indicated that the decision was made in goods space. Notably, the spatio-temporal dissociation between the offers and the presentation of action targets was crucial because it ensured that the neural activity encoding the choice outcome did not reflect a computation taking place in actions space. In the present experiments, we sought to undertake a similar approach while manipulating the action costs.

Second, varying the action costs introduced a significant challenge because the offer presentation had to instruct the subject (a rhesus monkey) about the action cost while preventing the animal from planning the action itself. We considered using a task in which actions would be reaching movements and action costs would be manipulated by resisting or assisting loads[18,19]. However, the biomechanics of the arm makes it difficult to dissociate the action cost from the spatial components of the movement. Specifically, two reaching movements of equal amplitude towards different directions generally bear different costs[20]. Thus, we designed a task in which animals chose between two juices offered in variable amounts. Offers were associated with radial eye movements in different directions, and different saccade amplitudes imposed variable action costs. We reasoned that if the initial fixation point is straight ahead of the subject, the action cost associated with an eye movement is essentially independent of the saccade direction (isotropic) and only depends on the saccade amplitude.

Our experimental design made it possible to examine the neuronal representation of goods and values in multiple frames of reference (commodity-based, action-based, cost-based) and to identify neuronal activity reflecting the action cost independently of an action plan. We report three primary results. First, as a population, neurons in OFC represent the identities and values of goods available for choice in two reference frames, namely commodity-based and cost-based. Second, we find neuronal activity encoding the choice outcome before presentation of the action targets, suggesting that decisions were made in goods space. Third, a population of neurons encoding the offer value reflects the integration of juice type, juice amount, and action cost. These results generalize previous observations on good-based decisions to choices under variable action costs, and demonstrate a remarkable degree of adaptability in the neural circuit underlying economic decisions.

## Results

**Experimental design and choice patterns**. Figure 1a illustrates the experimental design. Notably, each offer provided information about all the determinants of value (juice type, quantity, and action cost), while preventing the animal from planning the saccade necessary to obtain the offer. Behavioral evidence indicated that the experimental manipulation was effective. Figure 1b illustrates the choice pattern recorded in one representative session. Trials were divided in two groups depending on whether juice A was offered at low cost or at high cost. The gray sigmoid is displaced to the right, indicating that the relative value of juice A was higher when juice A was offered at low cost. This effect was relatively modest, but it was consistent across sessions for both monkeys. For a quantitative analysis of choice patterns, we constructed a logistic model that provided measures for the relative value of the two juices ($\rho$), the difference in action cost ($\xi$), the choice hysteresis related to the chosen juice ($\eta$) and to the chosen cost ($\varphi$), and the spatial biases related to the offer position ($\delta$) and to the target position ($\varepsilon$) (see Methods, Eq. 1).

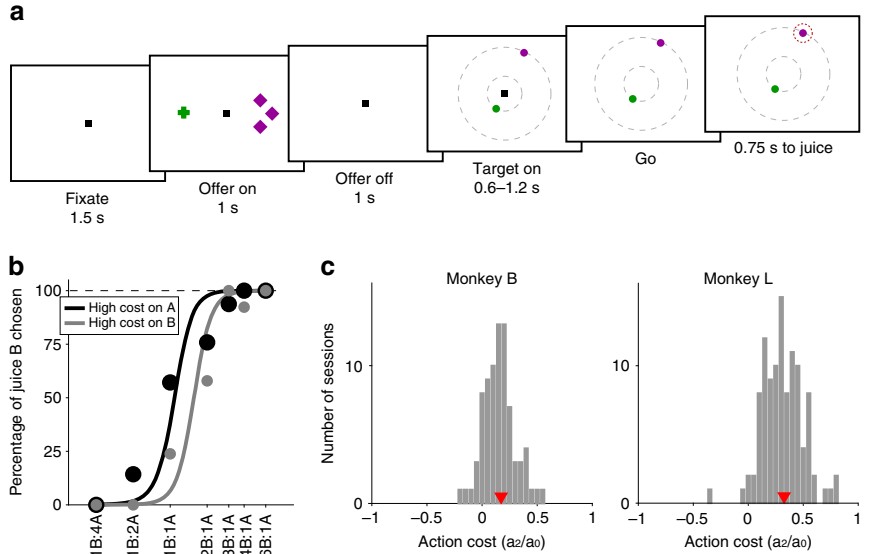

**Fig. 1** Experimental design and behavioral analysis. **a** At the beginning of the trial, the monkey fixated a center point on the monitor. After 1.5 s, two offers appeared to the left and right of the fixation point. The offers were represented by sets of color symbols, with the color indicating the juice type, the number of squares indicating juice amount and the shape of the symbols indicating the action cost associated with the offer (crosses, low cost; diamonds high cost). The offers remained on the monitor for 1 s, then disappeared. The monkey continued fixating the center point for another 1 s. At the end of this delay, two saccade targets (two color dots) appeared. The two saccade targets were located on two (invisible) concentric rings centered on the fixation point. The monkey maintained fixation for a randomly variable delay (0.6–1.2 s) before the center fixation point was extinguished (go signal), at which point the monkey indicated its choice with a saccade. Importantly, the association between the symbols (cross, diamond) and the saccade amplitudes (short, long) was known to the monkey. **b** Choice patterns, one session. The percentage of B choices is plotted against the ratio #B:#A, where #A and #B are quantities of juice A and B, respectively. Trials were separated in two groups depending on the action cost for juice A. The choice pattern obtained when juice A had a high cost (black) was displaced to the left (lower indifference point) compared with the choice pattern obtained when juice A had a low cost (gray). The regression lines were obtained with a simplified version of Eq. (1) in which terms $a_3$ to $a_6$ were removed. The action cost can be measured as $\xi = a_2/a_0$ (Eq. 1). **c** Distribution of action costs measured across sessions for monkey B (79 sessions, median($\xi$) = 0.174, $p < 10^{-10}$, Wilcoxon signed-rank test) and for monkey L (107 sessions, median($\xi$) = 0.327, $p < 10^{-10}$, Wilcoxon signed-rank test). Red triangles indicate median values

Pooling data from two animals, our data set included 223 behavioral sessions. In some sessions, the animal presented a significant bias in favor of saccade targets located in the left hemifield. This effect was quantified by the normalized coefficient $\varepsilon$ (Eq. 1), and the logistic analysis indicated that $\varepsilon$ was significantly different from zero in 36 sessions ($p < 0.01$). We interpret this target-related spatial bias as owing to rightwards saccades imposing an additional action cost to the animal. The ultimate reasons of this effect are not clear. However, since the location of the saccade targets was initially unknown to the animal, this spatial bias, when present, prevented us from addressing the question of interest in this study, namely whether decisions under known and variable action costs can take place in a non-spatial representation. Thus we excluded from the analysis sessions in which the spatial bias was statistically significant. Subsequent analyses focused on the remaining data set, which included 186 sessions.

The difference in action cost was quantified by the normalized coefficient $\xi$ (Eq. 1). We thus examined the distribution of $\xi$ across sessions. The difference in saccade amplitude had a significant effect on choices in both animals (monkey B, median ($\xi$) = 0.174, $p < 10^{-10}$; monkey L, median ($\xi$) = 0.327, $p < 10^{-10}$; Wilcoxon signed-rank test; Fig. 1c).

We previously observed that, other things equal, monkeys tend to choose on any given trial the same juice chosen (and received) in the previous trial[11]. This phenomenon, termed choice hysteresis, was quantified by the normalized coefficient $\eta$ (Eq. 1). Choice hysteresis was significantly present in our data set (median($\eta$) = 0.228, $p < 10^{-10}$; Wilcoxon signed-rank test). In other words, the effect of choosing juice B in previous trials was on average equivalent to adding 0.228 units to juice B in the

current trial. We also tested whether choices were affected by the cost incurred in previous trial. This effect, quantified by the normalized coefficient $\varphi$ (Eq. 1), was not significant across sessions (median($\varphi$) = $2.5 \times 10^{-4}$, $p = 0.64$; Wilcoxon signed-rank test). Finally, the normalized coefficient $\delta$ quantified offer-based spatial biases. Across sessions, this effect was statistically significant but rather small (median($\delta$) = 0.022, $p < 0.01$; Wilcoxon signed-rank test).

**Encoding of goods and values in multiple reference frames**. We recorded from 786 neurons in the central OFC of two monkeys (B, 367 cells; L, 419 cells). Firing rates were analyzed in nine time windows aligned with different behavioral events (see Methods). A "neuronal response" was defined as the activity of one neuron in one time window as a function of the trial type.

In principle, multiple frames of reference could be used to represent good identities and values in the present task. In a commodity-based reference frame, values would be attached to a specific juice type (or, equivalently, to a specific color); in a location-based reference frame, values would be attached to the spatial location of the offer; in a cost-based reference frame, values would be attached to the option with high or low cost (or, equivalently, to a specific symbol). A qualitative inspection indicated that neuronal responses typically did not depend on the spatial locations of the offers. Conversely, a significant fraction of neurons encoded the value of individual offers (offer value), the choice outcome and the chosen value. Surprisingly, some neurons appeared to encode these variables in a commodity-based reference frame, whereas other neurons appeared to encode these variables in a cost-based reference frame. Figure 2 illustrates

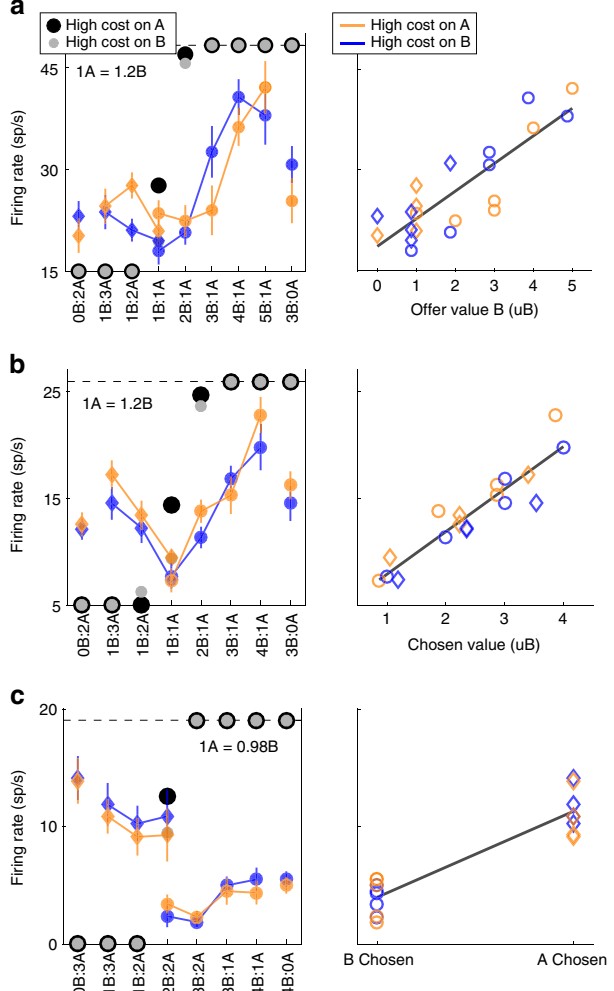

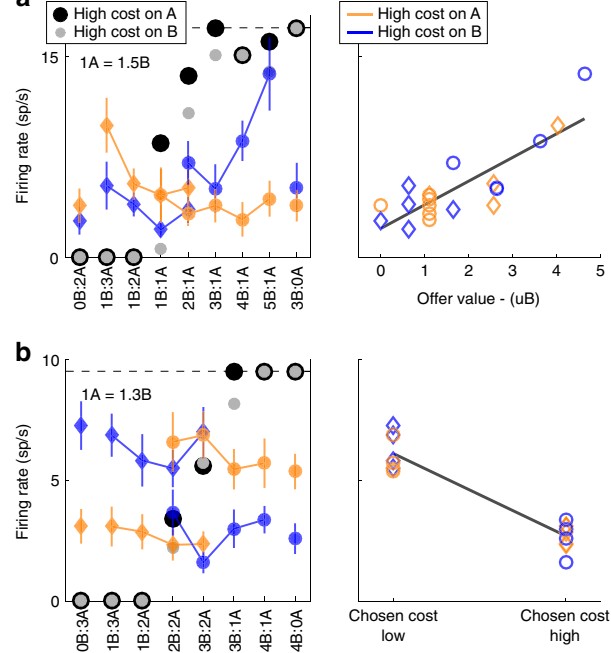

**Fig. 2** Encoding of decision variables in juice-based reference frame. **a** Neuronal response encoding the variable *offer value B* (post-offer time window). In the left panel, the x axis represents different offer types ranked by the ratio #B:#A. Black and gray symbols represent the percentage of B choices measured for A−:B+ trials and A+:B− trials, respectively. Color symbols represent the neuronal firing rate, with diamonds and circles, indicating trials in which the monkey chose juice A and juice B, respectively. Blue and orange indicate A+:B− trials and A−:B+ trials, respectively. Error bars indicate SEM. In the right panel, the same neuronal response is plotted against the variable *offer value B*. The black line is derived from a linear regression ($R^2 = 0.73$). **b** Response encoding the *chosen value* (late-delay time window). In the right panel, the response is plotted against the variable *chosen value* expressed in units of juice B. The black line is derived from a linear regression ($R^2 = 0.86$). **c** Response encoding the *chosen juice* (post-offer time window). In the right panel, the black line is derived from a linear regression ($R^2 = 0.85$). All conventions in **b** and **c** are as in **a**

**Fig. 3** Encoding of decision variables in cost-based reference frame. **a** Neuronal response encoding the variable *offer value −* (post-offer time window). In the right panel, the neuronal response is plotted against the variable *offer value −* expressed in units of juice B. The black line is derived from a linear regression ($R^2 = 0.69$). **b** Response encoding the variable *chosen cost* (pre-juice time window). In the right panel, the black line is derived from a linear regression ($R^2 = 0.88$). All conventions are as in Fig. 2a

a few examples. The response in Fig. 2a varied as a linear function of the value of juice B (variable *offer value B*). Similarly, the response in Fig. 2b varied as a linear function of the value chosen (variable *chosen value*). Finally, the response in Fig. 2c was roughly binary—high when the animal chose juice A and low when the animal chose juice B (variable *chosen juice*). For these three cells, the modulation due to the action costs appeared negligible. In contrast, the two responses depicted in Fig. 3 were primarily affected by the action cost. Specifically, the response in Fig. 3a varied as a linear function of the value associated with the high-cost offer (variable *offer value −*). The response in Fig. 3b

was roughly binary—high when the animal chose the low-cost option and low when the animal chose the high-cost option (variable *chosen cost*). Thus, neuronal representations in two reference frames seemed to coexist during this choice task.

For a statistical analysis, we proceeded in steps. First, we submitted each neuronal response to two three-way ANOVAs (factors (trial type × offer A location × target A location); factors (trial type × chosen offer location × chosen target location); see Methods and Table 1). We imposed a significance threshold $p < 0.001$. Responses that passed this criterion for at least one factor were identified as "task-related" and included in subsequent analyses. Confirming previous observations[4], many more neurons were modulated by the trial type (276 cells = 35%) compared with offer A location (35 cells = 4.5%), target A location (40 cells = 5.1%), chosen offer location (36 cells = 4.6%), or chosen target location (63 cells = 8.0%). Overall, 774 responses from 317 cells (40.3%) were modulated by at least one factor, and only these responses were included in subsequent analyses.

Next, we defined a large number of variables potentially encoded in OFC, and we used unbiased statistical procedures to identify a small subset of variables that best explained the neuronal data set. Previous work on decisions under fixed action costs already excluded numerous candidate variables in favor of variables *offer value*, *chosen value*, and *chosen juice*[4,21]. Building on these observations, here we focused on variables defined by (or disambiguated through) variable action costs. As noted above, the experimental design afforded multiple reference frames. Thus, we examined commodity-based variables *offer value (juice)* and *chosen juice*, cost-based variables *offer value (cost)* and *chosen cost*, location-based variables *offer value (location)* and *chosen location*, and action-based variables *offer value (target)* and *chosen*

**Table 1 Task-related responses in different time windows**

|            | Trial type | Offer A location | Target A location | Chosen offer location | Chosen target location |
|------------|-----------|------------------|-------------------|-----------------------|------------------------|
| Pre-offer  | 1   | 0  | 1  | 0  | 0  |
| Post-offer | 121 | 18 | 0  | 23 | 0  |
| Late-delay | 95  | 11 | 0  | 12 | 1  |
| Mem-delay  | 55  | 8  | 0  | 7  | 1  |
| Pre-target | 37  | 4  | 1  | 1  | 0  |
| Post-target| 65  | 3  | 22 | 1  | 12 |
| Pre-go     | 53  | 0  | 20 | 0  | 17 |
| Pre-juice  | 129 | 1  | 6  | 1  | 36 |
| Post-juice | 103 | 1  | 1  | 0  | 18 |
| At least 1 | 276 | 35 | 40 | 36 | 63 |

The table reports the results of two three-way ANOVAs (factors (trial type×offer A location×target A location); factors (trial type×chosen offer location×chosen target location). Each column represents one factor, each row represents one time window, and numbers represent the number of cells significantly modulated by the corresponding factor ($p < 0.001$). The bottom row indicates, for each factor, the number of cells that passed the criterion in at least one of the nine time windows. The factor trial type is common to the two ANOVAs

**Table 2 Defined variables**

|    | Collapsed variable      | Variable          | Definition                                                                  | Reference frame |
|----|-------------------------|-------------------|-----------------------------------------------------------------------------|-----------------|
| 1  | Offer value (juice)     | Offer value A     | $\rho$ #A $+ \xi\, \delta_{\text{juice A,}+}$                                | Commodity       |
| 2  |                         | Offer value B     | #B $+ \xi\, \delta_{\text{juice B,}+}$                                       | Commodity       |
| 3  |                         | Chosen juice      | 1 if juice B is chosen, 0 if juice A is chosen                              | Commodity       |
| 4  | Offer value (cost)      | Offer value $-$   | Offer value A if A is high-cost, offer value B if B is high-cost           | Cost            |
| 5  |                         | Offer value $+$   | Offer value A if A is low-cost, offer value B if B is low-cost             | Cost            |
| 6  |                         | Chosen cost       | 1 if low-cost offer is chosen, 0 if high-cost offer is chosen              | Cost            |
| 7  | Offer value (location)  | Offer value L     | Value of the juice offered on the left                                      | Visual          |
| 8  |                         | Offer value R     | Value of the juice offered on the right                                     | Visual          |
| 9  |                         | Chosen location   | 1 if left offer is chosen, 0 if right offer is chosen                      | Visual          |
| 10 | Offer value (target)    | Offer value target L | Value of the juice associated with target in the left hemifield          | Action          |
| 11 |                         | Offer value target R | Value of the juice associated with target in the right hemifield         | Action          |
| 12 |                         | Chosen target     | 1 if saccade to left hemifield, 0 if saccade to right hemifield           | Action          |
| 13 |                         | Cost of A         | 1 if offer A is low-cost, 0 if offer A is high-cost                        |                 |
| 14 |                         | Offer A location  | 1 if juice A is offered on left, 0 if juice A is offered on right          |                 |
| 15 |                         | Target A location | 1 if target A is in left hemifield, 0 if target A is in right hemifield    |                 |
| 16 |                         | Offer $+$ location | 1 if low-cost offer is on the left, 0 if low-cost offer is on the right    |                 |
| 17 |                         | Target $+$ location | 1 if low-cost target is in the left hemifield, 0 otherwise               |                 |
| 18 |                         | Spatial congruence | 1 if offers and targets are spatially congruent, 0 otherwise             |                 |
| 19 |                         | Chosen value      | Offer value A if juice A chosen, offer value B if juice B chosen           |                 |

In any given trial, #A and #B were, respectively, the quantities of juice A and juice B offered to the animal, $\rho$ was the relative value of the two juices, and $\xi$ was the action cost. Parameters $\rho$ and $\xi$ were obtained from the logistic regression (Eq. 1). The variable spatial congruence was set $= 1$ ($= 0$) if the offer and the saccade target associated with a given juice were presented in the same (opposite) hemifield

*target.* In addition, we defined variables that captured the association between different reference frames including the association between juice type and cost (*cost of A*), juice type and offer location (*offer A location*), juice type and target location (*target A location*), cost and offer location (*offer + location*), cost and target location (*target + location*), and offer location and target location (*spatial congruence*). Finally, we tested the variable *chosen value*. All the variables included in the analysis are defined in Table 2.

Each response that passed the analysis of variance criterion was regressed on each variable. A variable was said to "explain" the response if the regression slope differed significantly from zero ($p < 0.05$). Each linear regression also provided the $R^2$. For variables that did not explain the response, we set $R^2 = 0$. The variable with the largest $R^2$ provided the "best fit" for the neuronal response. Figure 4 illustrates the results obtained for the population. Figure 4a indicates the number of responses explained by each variable in each time window. Notably, each response could be explained by more than one variable and thus could contribute to multiple bins in this panel. Figure 4b illustrates a complementary account. Here, each response was assigned to the variable that provided the best fit. In early time windows, the dominant variables were *offer value*

*(juice)*, *offer value (cost)*, and *chosen value*. In late time windows, after target presentation and upon juice delivery, the dominant variables were *chosen value*, *chosen juice*, and *chosen cost*. Two procedures—stepwise and best-subset—were used to identify the variables that best explained the neuronal data set (see Methods). As in previous work[16], variables were selected separately for pre- and post-target time windows. For early time windows, both procedures selected variables *offer value (juice)*, *offer value (cost)*, and *chosen value*. For late time windows, both procedures selected variables *chosen value*, *chosen juice*, and *chosen cost*. Figure 5 illustrates the percentage of neurons encoding each of the selected variables across different time windows.

To summarize, neurons in OFC encoded the values of individual offers, the chosen value and the binary choice outcome. Remarkably, some neurons encoded the *offer value (juice)*, whereas other neurons encoded the *offer value (cost)*. Similarly, some neurons encoded the *chosen juice*, whereas other neurons encoded the *chosen cost*. Thus the offer value and the binary choice outcome were simultaneously represented in two reference frames (commodity-based and cost-based). This result demonstrates a high degree of flexibility in the decision circuit.

**a**

| | Offer value (juice) | Chosen juice | Offer value (cost) | Chosen cost | Offer value (location) | Chosen location | Offer value (target) | Chosen target | Cost of A | Offer A location | Target A location | Offer + location | Target + location | Spatial congruence | Chosen value |
|---|---|---|---|---|---|---|---|---|---|---|---|---|---|---|---|
| Post-offer | 100 | 75 | 73 | 37 | 67 | 29 | 32 | 2 | 21 | 24 | 1 | 6 | 1 | 1 | 102 |
| Late-delay | 76 | 50 | 54 | 24 | 39 | 14 | 20 | 0 | 25 | 11 | 0 | 7 | 1 | 6 | 65 |
| Mem-delay | 40 | 32 | 28 | 15 | 24 | 8 | 11 | 1 | 14 | 8 | 1 | 5 | 1 | 1 | 40 |
| Pre-target | 28 | 25 | 22 | 11 | 11 | 4 | 5 | 3 | 7 | 4 | 0 | 2 | 1 | 3 | 26 |
| Post-target | 41 | 37 | 34 | 24 | 12 | 2 | 24 | 15 | 35 | 4 | 21 | 2 | 17 | 3 | 41 |
| Pre-go | 31 | 34 | 31 | 33 | 7 | 2 | 12 | 9 | 23 | 1 | 17 | 1 | 20 | 2 | 26 |
| Pre-juice | 95 | 93 | 82 | 79 | 9 | 4 | 38 | 29 | 27 | 3 | 5 | 1 | 18 | 3 | 48 |
| Post-juice | 77 | 83 | 59 | 50 | 13 | 1 | 26 | 14 | 14 | 6 | 2 | 0 | 7 | 2 | 46 |

**b**

| | Offer value (juice) | Chosen juice | Offer value (cost) | Chosen cost | Offer value (location) | Chosen location | Offer value (target) | Chosen target | Cost of A | Offer A location | Target A location | Offer + location | Target + location | Spatial congruence | Chosen value |
|---|---|---|---|---|---|---|---|---|---|---|---|---|---|---|---|
| Post-offer | 42 | 11 | 19 | 5 | 9 | 5 | 0 | 0 | 2 | 4 | 0 | 0 | 0 | 0 | 37 |
| Late-delay | 33 | 7 | 11 | 1 | 5 | 2 | 0 | 0 | 3 | 3 | 0 | 2 | 0 | 0 | 32 |
| Mem-delay | 19 | 3 | 6 | 2 | 5 | 1 | 0 | 0 | 3 | 4 | 0 | 0 | 0 | 0 | 19 |
| Pre-target | 12 | 5 | 6 | 3 | 1 | 0 | 0 | 0 | 2 | 3 | 0 | 0 | 0 | 1 | 9 |
| Post-target | 8 | 11 | 3 | 7 | 0 | 0 | 3 | 4 | 10 | 1 | 10 | 0 | 5 | 0 | 20 |
| Pre-go | 7 | 14 | 3 | 14 | 0 | 0 | 3 | 1 | 10 | 0 | 7 | 0 | 4 | 0 | 9 |
| Pre-juice | 6 | 64 | 8 | 45 | 1 | 0 | 2 | 6 | 3 | 3 | 1 | 0 | 2 | 1 | 7 |
| Post-juice | 8 | 52 | 4 | 29 | 0 | 0 | 1 | 5 | 1 | 1 | 1 | 0 | 3 | 1 | 14 |

**Fig. 4** Population summary of linear regressions (all time windows). **a** Explained responses. Row and columns represent time windows and variables, respectively. In each location, the number indicates the number of responses explained by the corresponding variable in that time window. For example, in the post-offer time window, the variable *offer value (juice)* explained 100 responses. The same numbers are also represented in gray scale. Note that each response could be explained by more than one variable and thus could contribute to multiple bins in this panel. **b** Best fit. In each location, the number indicates the number of responses for which the corresponding variable provided the best fit (highest $R^2$). For example, in the post-offer time window, the variable *offer value (juice)* provided the best fit for 42 responses. The numerical values are also represented in gray scale. In this plot, each response contributes to at most one bin. Qualitatively, *offer value (juice)*, and *chosen value* seem to be the dominant variables in early time windows. Conversely, *chosen juice* and *chosen cost* seem to be the dominant variables in late time windows. The variable *chosen value* is present througout the trial

**Good-based economic decisions under variable action costs.** We next examined whether decisions in our task were made before target presentation and thus in goods space. As described above, two groups of neurons—namely *chosen juice* and *chosen cost*—reflected the binary choice outcome. Both variables were most prominent late in the trial, before, and after juice delivery. However, both variables also provided the best fit for a sizeable number of cells in early time windows, shortly after the offer (Fig. 4). Thus, the critical question was whether the decision of the animal could be reliably predicted from the neuronal activity recorded before target presentation. We examined these two signals in turn.

First, we identified *chosen juice* cells (largest sum($R^2$) across all time windows; 108 cells)[11]. We thus examined the activity of these neurons in early time windows. For each cell, we refer to the juice eliciting higher firing rates as the "encoded" juice (juice E), and to the juice eliciting lower firing rates as the "other" juice (juice O). We divided trials depending on the chosen juice and we

examined the activity profile for the population (Fig. 6a). The two traces were clearly separated starting ~ 250 ms after the offer and throughout the delay, indicating that decisions were completed long before action planning.

Choices in our experiments reliably depended on the saccade amplitude, but the behavioral effect was relatively small (Fig. 1c). Thus, it is conceivable that upon easy decisions, when one value clearly dominated, monkeys effectively ignored the difference in action cost. One concern was whether the effect illustrated in Fig. 6a was driven by trials in which animals ignored the action cost. To address this issue, we defined "cost-overt" offer types as those in which the animal chose the low-cost offer more frequently ( > 10%) than the high-cost offer, conditioned on the animal choosing each option at least twice (e.g., Fig. 1b, 1B:1 A). Conversely, "cost-covert" offer types were those in which the animal consistently chose the same option independently of the action cost (e.g., Fig. 1b, 1B:4A). We divided trials in four groups depending on the chosen juice (E or O) and on whether the effect

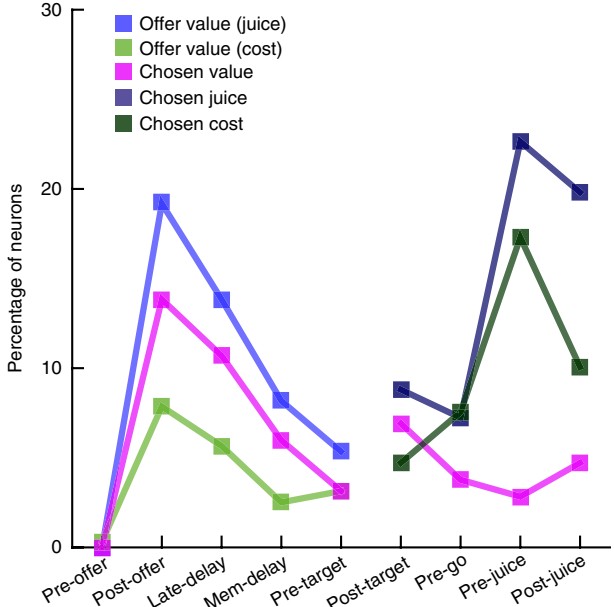

**Fig. 5** Time course of encoded variables. In total, 317 cells were task-related (see Methods and Table 1). The y axis in this figure represents the percentage of these neurons encoding the corresponding variable in each time window

of action costs was overt (o) or covert (c). For each group, we averaged the activity profiles across trials and across cells. Visual inspection of Fig. 6b indicates that the two traces for covert action costs—Ec and Oc trials—were clearly separated. Crucially, the two traces for overt action costs—Eo and Oo trials—were also separated. We quantified this separation using a receiver operating characteristic (ROC) analysis, which returned the area under the curve (AUC; see Methods). For each cell, AUC > 0.5 (< 0.5) indicated higher (lower) firing rates when the animal chose the juice encoded by the cell (Eo trials). For example, for the cell in Fig. 2c, AUC = 0.79. In other words, the activity of this neuron recorded prior to target presentation reliably revealed the eventual choice outcome. Across the population of *chosen juice* cells, the distribution of AUC was significantly displaced compared with chance level in each of three non-overlapping time windows (Fig. 6c), confirming that cost-affected decisions were made prior to target presentation.

We conducted a similar analysis on the population of neurons encoding the *chosen cost* (62 cells). Again, we identified these neurons based on their activity across all time windows and we analyzed the activity preceding target presentation. We first considered all the trials, and divided them depending on whether the animal chose the high-cost or the low-cost offer. In this analysis, the two traces separated starting ~ 250 ms after the offer (not shown). We then focused on cost-overt offer types and we conducted an ROC analysis. Across the population, the distribution of AUC was significantly above chance level in two of three non-overlapping time windows (Fig. 6d).

In summary, both *chosen juice* and *chosen cost* neurons revealed the binary choice outcome well before target presentation, indicating that decisions were made in goods space.

**Integration of action costs and other determinants of value.** Thus far, we have shown (1) that decisions were informed by the action cost, (2) that decisions were good-based, and (3) that three groups of neurons encoded variables *offer value (juice)*, *offer value*

(cost), and *chosen value* cells. Next we examined whether these value-encoding cells integrated all three determinants affecting choices (i.e., juice type, quantity, and action cost).

First, we examined neurons encoding the *offer value (juice)*. These cells were associated to a juice type and their firing rates varied as a function of the juice quantity. The question is whether firing rates also varied as a function of the action cost. In principle, this issue could be addressed by defining two variants of each *offer value (juice)* variable—one cost-affected and one cost-independent. Both variants could be included in the variable selection analysis and the variant with the higher explanatory power would be selected. However, because the difference in action cost was relatively modest in our task, the two variants were quantitatively close and the variable selection analysis did not disambiguate between them. We thus proceeded as follows.

The variable selection analysis described in the previous section was performed using cost-affected variables (Table 2). However, we repeated the analysis using cost-independent variables and we obtained identical results (same selected variables). For each response, we computed the difference between the $R^2$ ($\Delta R^2$) obtained with the two variables (cost-affected and cost-independent; see Methods). Across the population, the distribution of $\Delta R^2$ was tendentially displaced toward negative values (mean $(\Delta R^2) = -0.0065$, $p = 0.055$; Fig. 7a). In other words, neurons encoding the *offer value (juice)* did not seem to integrate action costs with the other determinants of value. To further quantify the effects of action cost, we regressed each *offer value (juice)* response against the offer value using an analysis of covariance (ANCOVA, parallel model) and grouping data by the action cost. Although the factor action cost was statistically significant for 25% of responses, there was no consistent correlation between the slope of the encoding and the sign of this effect (Supplementary Fig. 1a). Thus, we did not find a systematic effect of the action cost for this population.

We conducted a similar analysis on neurons encoding the *offer value (cost)*. In this case, each response was associated to a cost level and firing rates varied as a function of the juice quantity. Thus, we examined whether firing rates also varied as a function of the juice type. Again, we defined two variants of the variable— one commodity-affected and one commodity-independent—and we confirmed that the variable selection analysis yielded the same results for both variants. For each response, we computed the difference between the two $R^2$ obtained with the two variables ($\Delta R^2$). Across the population, $\Delta R^2$ was significantly > 0 (mean $(\Delta R^2) = 0.017$, $p < 0.05$; Fig. 7b). Hence, these neurons encoded a value variable integrating juice type and quantity in a cost-based representation. An ANCOVA confirmed this finding (see Methods; Supplementary Fig. 1b).

Finally, we examined *chosen value* cells. Their activity depended on both the juice type and the juice quantity; we assessed whether it also reflected the action cost. We defined two variants of the variable and we confirmed that the variable selection analysis yielded the same results. For each response, we computed the difference in $R^2$. Interestingly, we found a dissociation between early and late time windows. In the post-offer time window, the distribution of $\Delta R^2$ was significantly < 0 (mean $(\Delta R^2) = -0.013$, $p < 0.01$; Fig. 7c). Conversely, in the post-target time window, the distribution of $\Delta R^2$ was tendentially > 0 (mean $(\Delta R^2) = 0.0089$, $p = 0.058$; Fig. 7c). The difference across time windows was statistically significant (mean $\Delta(\Delta R^2) = 0.022$, $p < 0.001$; Fig. 7c). Thus, *chosen value* responses progressed from integrating only juice type and quantity to integrating all three determinants. An ANCOVA confirmed these observations (Supplementary Fig. 1c, d). Of note, *chosen value* responses in different time windows often came from different cells (chi-square test, $p = 0.14$).

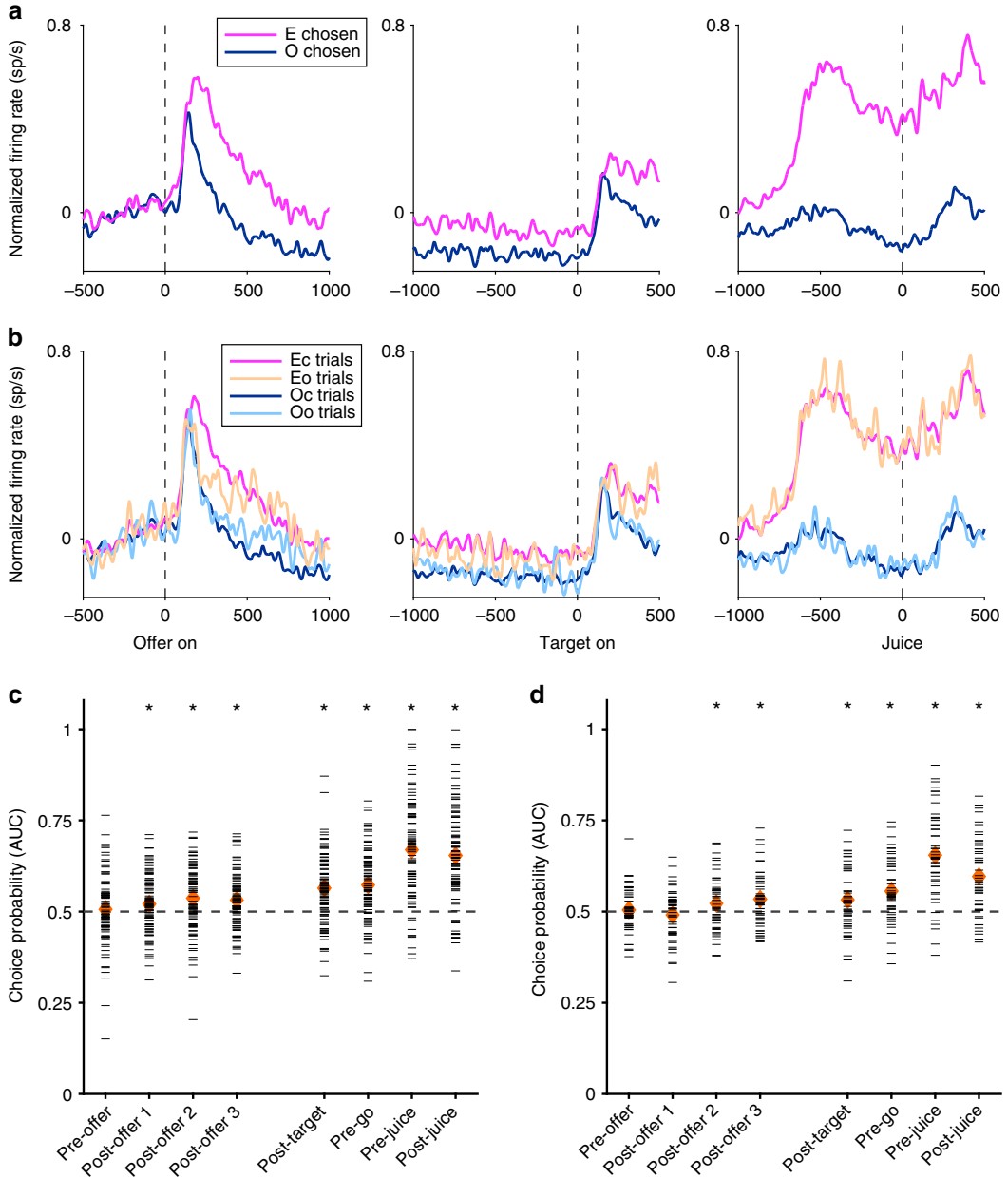

**Fig. 6** Population activity profiles and ROC analysis for cells encoding the choice outcome. **a** *Chosen juice* cells ($N = 108$). Neurons were classified across all time windows. Activity profiles were aligned at offer on (left panel), target on (center panel), and juice delivery (right panel). Trials were divided depending on whether the animal chose the juice encoded by the cell (E) or the other juice (O). The two traces clearly separated within 250 ms after the offer. **b** *Chosen juice* cells ($N = 97$). Trials were divided depending on whether the animal chose the juice encoded by the cell (E) or the other juice (O) and on whether the effect of different action costs was covert (c) or overt (o). Average traces shown here are from the cells for which we could compute all four traces ($\geq 2$ trials per trace). Notably, the activity profile in Eo trials is elevated compared with that in Oo trials. **c** ROC analysis for 97 *chosen juice* cells in cost-overt trials (same population as in **b**). Each horizontal gray line represents one neuronal response. Each orange diamond represents the mean AUC, which was significantly $> 0.5$ in each of three non-overlapping time windows before target presentation (post-offer 1, mean(AUC) = 0.521, $p < 0.02$; post-offer 2, mean(AUC) = 0.537, $p < 0.0005$; post-offer 3, mean(AUC) = 0.532, $p < 0.0005$; $t$ test; see Methods). **d** ROC analysis for 51 *chosen cost* cells in cost-overt trials. Same convention as in **c**. Across the population, mean(AUC) was significantly $p > 0.5$ in two of three time windows before target presentation (post-offer 2 (mean(AUC) = 0.522, $p < 0.05$); post-offer 3 (mean(AUC) = 0.534, $p < 0.002$; $t$ test). c and d, an asterisk (*) indicates $p < 0.02$, $t$ test. AUC area under the curve, ROC receiver operating characteristic

## Discussion

In the experiments described here, options varied along three dimensions—juice type, quantity, and action cost. Importantly, the task design dissociated in time and space offer presentation from action planning. We reported three primary results. First, the neuronal population represented good identities and values in two reference frames, namely commodity-based and cost-based. Second,

a group of cells encoded values as an integrated quantity reflecting all the dimensions relevant to the decision. Third, neuronal activity encoding the choice outcome before presentation of the saccade target indicated that decisions (i.e., value comparisons) were made in goods space. We discuss these findings in reverse order.

Current views hold that decisions between goods take place in a non-spatial representation[6–9]. Conversely, decisions underlying

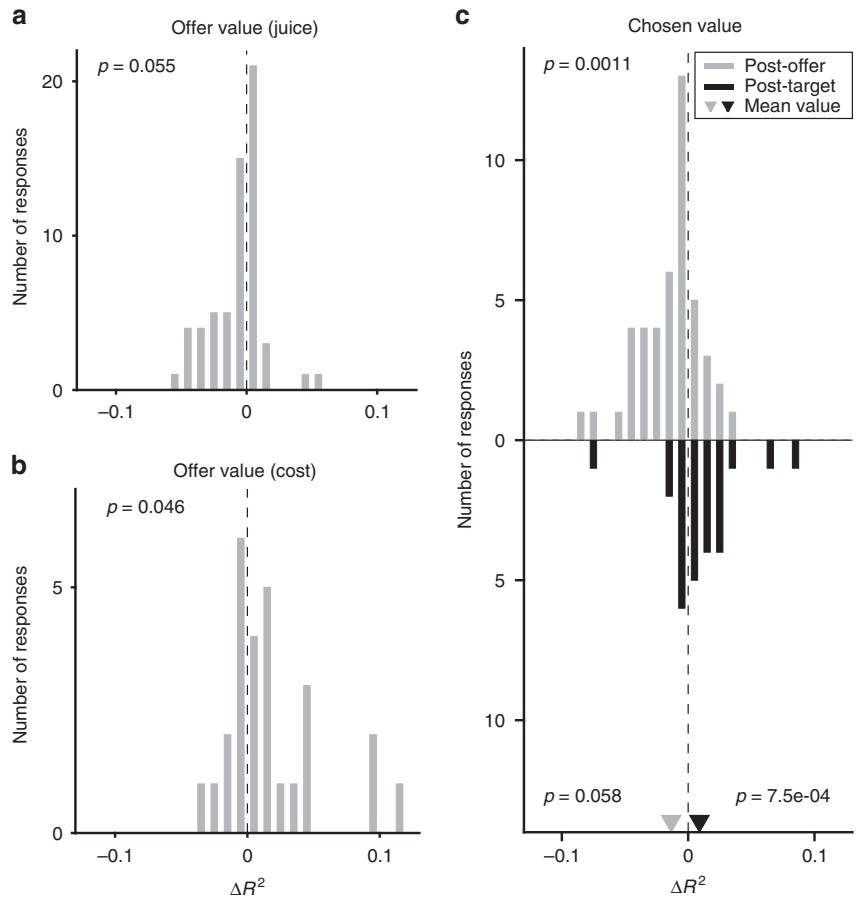

**Fig. 7** Model comparisons. **a** *Offer value (juice)* responses ($N = 60$). The x axis represents the difference $\Delta R^2$. Across the population, mean($\Delta R^2$) = $-0.0065$ ($p = 0.055$, Wilcoxon signed-rank test). **b** *Offer value (cost)* responses ($N = 27$). The x axis represents the difference $\Delta R^2$. Across the population, mean($\Delta R^2$) = 0.017 ($p < 0.05$, Wilcoxon signed-rank test). **c** *Chosen value* responses. The x axis represents the difference $\Delta R^2$. In this case, we examined separately early time windows (top, $N = 45$ responses) and late time windows (bottom, $N = 25$ responses). In early time windows, mean($\Delta R^2$) = $-0.013$ ($p < 0.01$, Wilcoxon signed-rank test). In late time windows, mean($\Delta R^2$) = 0.0089 ($p = 0.058$, Wilcoxon signed-rank test). The difference between the two measures was statistically significant ($p < 0.001$, Wilcoxon ranksum test)

action selection involve motor systems. Inspired by this observation, several authors argued that some value-based decisions take place in an action-based representation[6–8]. According to a unifying proposal, decisions generally emerge from multiple competitions taking place in parallel within and across brain regions and neuronal representations. In this view, good-based decisions and action-based decisions are particular cases of distributed-consensus mechanisms[6,22–24]. Critically, the distributed-consensus framework does not specify under what conditions a decision can be conceptualized as a decision between goods. In this respect, a particularly interesting case is that of choices between options that differ for their action cost. According to one proposal, these decisions are necessarily action-based[6,10]. Alternatively, action costs could be integrated with other determinants of value in a non-spatial representation. If so, decisions under variable action costs may in fact be good-based[3].

To establish whether a decision is good-based, one must dissociate offer presentation from action planning. If this condition is met, neural signals encoding the choice outcome prior to action planning reveal that the decision took place in goods space. Previous work used this approach to show that decisions between edible goods under fixed action cost are indeed good-based[16,17]. The present study extends earlier results to choices under variable action costs. In other words, our results indicate that decisions can be good-based even when the value of each good depends on

the effort necessary to obtain that good. Importantly, our findings are not inconsistent with the idea that motor regions play a primary role in the calculation of action costs. For example, previous work suggested a specific role of the anterior cingulate cortex (ACC) in choice under variable action cost[25–27]. Signals from ACC and/or other motor regions could provide an input for the computation of subjective values in OFC. Furthermore, our results do not exclude that in different conditions—e.g., when offer presentation and action planning are not dissociated—motor systems may participate in value comparison[6,9,28].

Previous studies argued that motor regions can represent multiple action plans at once[29,30]. Thus, one concern might be whether our animals made a decision early in the trial in actions space. Several considerations diffuse this concern. First, evidence for the simultaneous representation of multiple action plans is not conclusive. Indeed, responses that seem bimodal when averaged across trials might really be unimodal on any given trial[31]. One study that recorded from many cells simultaneously and conducted single-trial analyses concluded that neurons in premotor cortex process only one action plan at the time[31]. Furthermore, "bimodal" neurons were found in rostral F2 and in F7[29,32,33]. However, F7 is a prefrontal (not a motor) area[34,35] and neurons in rostral F2 are often associated with eye movements rather than arm movements[36,37]. Looking forward, it will be interesting to record from motor regions using our task, and to compare the

time course of neural activity across areas. For now, it seems safe to assume that action-based decisions in our task would necessarily take place after target presentation.

Economic choices depend on a variety of determinants (dimensions) along which goods may vary, and numerous studies found that neurons in the OFC integrate multiple determinants of value[4,21,38]. Most relevant here, Kennerley et al.[39] found neurons whose activity was modulated by the juice quantity, the probability and the action cost. The fraction of neurons presenting all three modulations at once was relatively small, but Kennerley's analysis was conservative and likely underestimated the degree of integration (owing to type II errors). In another study, Morrison and Salzman[40] showed that the same OFC neurons responded both to positive reinforcers (drops of juice) and negative reinforcers (air puffs). More recently, Hirokawa et al.[41] found in the rodent OFC neurons integrating reward quantity and decision confidence in a value signal. Last but not least, numerous studies found evidence for integration in the BOLD signal using a variety of behavioral manipulations[42,43]. Hence, the observation that *offer value* cells integrate juice type, juice quantity, and action costs is in line with established concepts. As for *chosen value* cells, we found evidence for full integration in late but not in early time windows. In general, the role played by these neurons in the decision process remains unclear. Future work should further examine this important issue in the light of current observations.

In spite of the robust evidence for dimensional integration in OFC, two previous reports reached different conclusions. In one study, the choice task varied juice quantity and action cost, or juice quantity, and time delay[44]. The authors found only few OFC neurons that integrated determinants in a value signal. However, their statistical procedures effectively fractionated the population of value-encoding cells, which likely impeded the identification of these neurons[21]. More recently, Blanchard et al. used a choice task varying juice quantity and the time at which information about the trial outcome was made available. Although both dimensions affected choice, the authors failed to find evidence for dimensional integration in OFC[15]. As discussed below, this result might reflect the failure to consider alternative reference frames.

Choice tasks often afford multiple reference frames. For example, in experiments that involve multiple commodities[45,46], goods and values may be represented in a commodity-based frame. Conversely, in tasks that involve a single commodity[44,47], goods and values are necessarily represented in some other reference frame. In choice tasks where options differ by a particular determinant—probability[21], delay[48], cost, etc.—a valid reference frame may be that defined by that determinant. In general, any characteristics of the choice task can provide a valid reference frame. For example, tasks in which options are presented sequentially[15,49,50] afford an order-based reference frame; tasks in which options are defined by the outcome of the previous trial[12,51] afford the corresponding reference frame; etc. In our experiment, valid frames included that defined by the commodity, the action cost, the spatial location of the offer and the saccade targets. Interestingly, different groups of cells represented goods and values in different frames. This result indicates that the representation of goods and values is highly malleable, and suggests that this neural circuit can reconfigure itself depending on the demands of the choice task. Of course, such malleability is a hallmark of prefrontal regions[52].

The fact that goods and values may be represented in multiple reference frames has far-reaching implications. To dissect the neural circuit of economic decisions, and specifically to establish what variables are encoded or not-encoded in a particular brain region during a choice task, it is generally necessary to examine multiple reference frames and to identify the reference frame that best accounts for the neuronal data. Conversely, failure to

consider a valid reference frame may explain some otherwise puzzling results. For example, in the study by Blanchard et al., monkeys chose between two options. Each option was a gamble with fixed probability, and the two options varied for the juice quantity and the informativeness[15]. Valid reference frames included that defined by the temporal order (order-based) and that defined by the informativeness (information-based). Neuronal recordings were performed in OFC. Focusing on the time window following the first offer, the authors regressed firing rates against the juice quantity and, separately, against the informativeness of the first offer. For each regressor, a sizeable fraction of cells had a significant effect, indicating that both determinants of value were represented in OFC. However, there was no correlation between the regression coefficients obtained for the two variables. The authors concluded that the neuronal population did not encode an integrated value. Of note, this claim was based on a negative result. Thus any source of noise—poor isolation, low cell count, etc.—effectively worked in favor of the conclusion. Most relevant here, the analysis assumed an order-based representation. In such reference frame, cells encoding the offer value of the first option would integrate the quantity and the informativeness of that option, and the two regression coefficients should indeed be correlated. However, this prediction does not hold in a different reference frame. In particular, if the representation was information-based, one group of cells would be associated with the informative option and another group of cells would be associated with the non-informative option. The regression coefficients obtained for quantity and informativeness might still be related in any given cell. However, once different groups of cells (including cells with positive and negative encoding) had been pooled together, there would be no systematic relation between regression coefficients. Importantly, other results of the Blanchard study seem inconsistent with an order-based representation (their Fig. 4a, b) and suggest an information-based representation (their Fig. 5a, b). Additional analyses[53] did not address this issue.

To conclude, assessing what reference frame(s) the brain adopts in any given condition is critical to understand the neuronal computations underlying the decision process. Together with other studies, our current findings indicate that the neural circuit underlying good-based decisions can reconfigure itself depending on the demands of the choice task. This reconfiguration may be conceptualized as a discrete form of context adaptation[9]. Importantly, the rules governing such adaptation and dictating the reference frame adopted on a given context are unclear and should be examined in future research.

## Methods

**Experimental procedures**. All experimental procedures conformed to the NIH Guide for the Care and Use of Laboratory Animals and were approved by the Institutional Animal Care and Use Committee (IACUC) at Washington University. Two rhesus monkeys (B, male, 9.0 kg; L, female, 6.5 kg) took part in the experiments. Before training, a head-restraining device and an oval recording chamber were implanted on the skull under general anesthesia, as previously described[16]. The behavioral task was controlled through a custom-written software based on Matlab (MathWorks) and freely available at http://www.monkeylogic.net/. In each session, an animal chose between two juices offered in different amounts and at variable action cost. Figure 1 illustrates the task design. At the beginning of each trial, the monkey fixated a point in the center of the monitor, within a tolerance window of 2° (in a subset of sessions the tolerance window was 3°). After 1.5 s, two offers appeared to the left and right of the fixation point. The offers were represented by sets of color symbols, with the color indicating the juice type, the number of symbols indicating juice amount, and the shape of the symbols indicating the action cost (cross for low cost; diamond for high cost). Different sets of juices were used across sessions. The offers remained on the monitor for 1 s. The monkey continued fixating for another 1 s, after which two saccade targets appeared. The two saccade targets, represented by two color dots corresponding to the color of the two juices, were located on two concentric rings centered on the fixation point. The radius for low-cost targets was 3.5°–4°; the radius for high-cost targets was 10°–16°. In each trial, one of the saccade targets was placed on the low-cost (small radius)

ring, whereas the other saccade target was placed on high-cost (large radius) ring. The two targets were always placed on opposite sides of the center fixation. The angle defining their position was selected on every trial among four possible values, corresponding to 22.5°, 157.5°, 202.5°, and 337.5° from azimuth. Thus, for each juice there were eight possible saccade target positions (two distances × four angles). The monkey maintained center fixation for a randomly variable delay (0.6–1.2 s), at the end of which the fixation point was extinguished (go signal). At that point the animal was allowed to indicate its choice with a saccade. The animal had to maintain peripheral fixation for an additional 0.75 s, at the end of which the chosen juice was delivered. In each session, the two juice quantities varied pseudo-randomly from trial to trial. The spatial positions of the offers, the action costs and the angle of the saccade targets varied pseudo-randomly and were counter-balanced across trials.

**Neuronal recordings**. Procedures for surgery, neuronal recordings, and spike sorting were similar to those described previously[16]. In brief, the recording chamber (main axes, 50 × 30 mm) was centered on stereotaxic coordinates (A30, L0), with the longer axis parallel to the coronal plane. During the experiments, animals sat in an electrically insulated enclosure (Crist Instruments) with their head restrained. The eye position was monitored with an infrared video camera (Eyelink; SR Research). Neuronal recordings were guided by structural MRI obtained for each animal before and after the implant, and focused on a region roughly corresponding to area 13 m[54]. In monkey B, we recorded from both hemispheres and recording locations ranged A36–A39 in the anterior–posterior direction (with the corpus callosum extending anteriorly to A36). In monkey L, we recorded from the left hemisphere and recording locations ranged A31–A36 in the anterior–posterior direction (with the corpus callosum extending anteriorly to A31). Tungsten electrodes (125 µm diameter, FHC) were advanced using custom-built motorized micro-drives, with a 2.5 µm resolution. We typically used four electrodes in each session. Electrical signals were amplified and band-passed filtered (high pass: 300 Hz, low pass: 6 kHz; Lynx 8, Neuralynx, Inc.). Action potentials were detected on-line and waveforms were saved to disk (25 kHz sampling rate; Power 1401, Spike 2; Cambridge Electronic Design). Spike sorting was conducted off-line (Spike 2; Cambridge Electronic Design) and only cells that appeared well isolated and stable throughout the session were included in the analysis.

**Analysis of choice patterns**. All analyses were conducted in Matlab (Math-Works). On any given trial, one "offer" was defined by a juice type, its quantity and its action cost (e.g., 3B−). An "offer type" was defined by two offers (e.g., [1 A + :3B−]). In this notation, "−" indicates high action cost (long saccade) and " + " indicates low action cost (short saccade). A "trial type" was defined by two offers and a choice (e.g., [1 A + :3B−, A]). Of note, the position of each saccade target was defined by a distance (two possible values) and an angle (four possible values). For the purpose of all the analyses, we turned the angle into a binary variable corresponding to whether the target associated to juice A or the chosen target was placed in the left hemisphere or in the right hemisphere.

In the behavioral analysis, we examined several factors that could affect choices, including the juice quantity, the action cost, the outcome of the previous trial (choice hysteresis), a term capturing a visual side bias, and a term capturing a saccade side bias. We thus constructed the following logistic model:

$$\text{choice B} = 1/(1 + e^{-X})$$
$$X = a_0 \#B - a_1 \#A + a_2 \left( \delta_{\text{juice B},+} - \delta_{\text{juice A},+} \right)$$
$$+ a_3 \left( \delta_{n-1, B} - \delta_{n-1, A} \right) +$$
$$+ a_4 \left( \delta_{\text{cost of B,cost } n-1} - \delta_{\text{cost of A,cost } n-1} \right)$$
$$+ a_5 \left( \delta_{\text{offer B,left}} - \delta_{\text{offer A,left}} \right) + a_6 \left( \delta_{\text{target B, left}} - \delta_{\text{target A, left}} \right)$$

(1)

where choice B = 1 if the animal chose juice B and 0 otherwise; #J was the quantity of juice J offered (with J = A, B); $\delta_{\text{juice J, +}} = 1$ if juice J was offered at low cost and 0 otherwise; $\delta_{n-1, J} = 1$ if in the previous trial the animal had chosen and received juice J and 0 otherwise; $\delta_{\text{cost of J, cost } n-1} = 1$ if the cost of J is the same as that chosen in the previous trial and 0 otherwise; $\delta_{\text{offer J, left}} = 1$ if the offer of juice J was placed to the left of the center fixation and 0 otherwise; and $\delta_{\text{target J, left}} = 1$ if the saccade target associated with juice J was placed in the left hemisphere and 0 otherwise. For each session, the logistic regression provided a measure for the relative value of the two juices ($\rho = a_1/a_0$), for the difference in action cost ($\xi = a_2/a_0$), for the choice hysteresis related to the chosen juice ($\eta = a_3/a_0$) and to the chosen cost ($\varphi = a_4/a_0$), and for the spatial biases related to the offer position ($\delta = a_5/a_0$) and to the target position ($\varepsilon = a_6/a_0$). In this formulation, each factor (action cost, hysteresis, spatial biases) is quantified as a value term, and all values are expressed in units of juice B. The relative value ($\rho$) is essentially the quantity of juice B that, when offered against 1 A, makes the animal indifferent between the two juices. The factor $a_1$ can be thought of as an inverse temperature capturing the steepness of the sigmoid once all the effects included in the logistic regression are accounted for.

**Task-related responses**. The neuronal analysis focused on sessions with no significant spatial bias (see Results). Each cell was analyzed in relation to the choice pattern recorded in the same session. In each trial, the neuronal activity was analyzed in nine time windows aligned with different behavioral events: pre-offer (0.5 s before the offer), post-offer (0.5 s after offer on), late-delay (0.5–1.0 s after offer on), mem-delay (0–0.5 s after offer off), pre-target (0.5 s before target on), post-target (0.5 s after target on), pre-go (0.5 s before the 'go'), pre-juice (0.5 s before the juice), and post-juice (0.5 s after the juice).

To identify task-related responses, each neuronal response was submitted to two three-way ANOVAs (factors [trial type × offer A location × target A location]; factors [trial type × chosen offer location × chosen target location]). We imposed a significance threshold $p < 0.001$. Responses that passed this criterion for at least one factor were identified as "task-related" and included in subsequent analyses.

**Variable selection analysis**. We conducted a series of analyses to identify the variables encoded by the neuronal population adopting the same general approach used in previous work[4,55]. We defined a large number of variables that neurons in the OFC might conceivably encode, and we applied procedures for variable selection to identify a small subset of variables that best explained the neuronal data set. Previous studies on decisions under fixed action costs already examined a large number of variables[4]. The results indicated that neurons in OFC encoded variables *offer value*, *chosen value*, and *chosen juice*. In contrast, the marginal explanatory power of other tested variables—including *total value*, *other value*, *value difference*, etc.—was very low. These results were replicated several times[16,56]. Thus, in the present study we focused on variables that were not already excluded in previous work, and in particular on variables defined by (or disambiguated through) variable action costs.

We were particularly interested in contrasting variables defined in different reference frames, and specifically variables defined by the juice type (commodity), the action cost, the spatial configuration of the offers and the spatial component of the action. For all reference frames, we expressed values in units of juice B (see above and Eq. 1). For the commodity-based frame, we defined offer value variables *offer value A* = $\rho$ #A + $\xi$ $\delta_{\text{juice A},+}$ and *offer value B* = #B + $\xi$ $\delta_{\text{juice B},+}$, where $\delta_{\text{juice J},+} = 1$ if juice J is low-cost and 0 otherwise, and J = A, B. Notably, each of these variables was associated with a specific juice type, and both variables reflected value as an integrated quantity: *offer value A* reflected the juice quantity (#A), the relative value of the two juices ($\rho$), and the action cost ($\xi$); *offer value B* reflected the juice quantity (#B) and the action cost ($\xi$). As in previous studies, we also defined the "collapsed" variable *offer value (juice)*, to which we assigned the higher of the two $R^2$ obtained for *offer value A* and *offer value B*. Similarly, for each of the other reference frames, we defined two offer value variables that reflected value as an integrated quantity. For example, the variable *offer value +* was associated to the low-cost offer and reflected the juice type, the juice quantity, and the relative value of the two juices (see Table 2). For each reference frame, we defined a collapsed offer value variable and a variable capturing the binary choice outcome. In addition, we defined variables that captured the association between different reference frames (see Results) and the variable *chosen value*. All the variables included in the analysis are defined in Table 2.

For details on the procedures used in the variable selection analysis, we refer to previous reports[4,16]. In brief, we examined collapsed offer value variables. As the task transitioned from choice to action at the time of target presentation, we performed the variable selection analyses separately for pre- and post-target time windows. Two procedures—stepwise and best-subset—identified a small number of variables that best explained the neuronal data set. In the stepwise procedure, we selected at each step the variable that provided the highest number of best fits within any time window. We then removed from the data set all the responses explained by this variable and we repeated the procedure on the residual data. The procedure was repeated until when the marginal explanatory power of any additional variable fell < 5%. In the best-subset procedure, we identified for n = 1, 2, 3,… the subset of *n* variables that collectively provided the highest explanatory power. Importantly, the best-subset procedure warrants optimality and the two procedures applied to our data set provided identical results.

**Activity profiles and ROC analysis**. Several analyses were conducted dividing trials in four groups, depending on the choice of the animal and on whether the offer type was cost-overt or cost-covert. For neurons encoding the *chosen juice*, trials were divided depending on whether the animal chose the juice encoded by the cell (juice E, defined as that which elicited higher activity) or the other juice (juice O). For neurons encoding the *chosen cost*, trials were divided depending on whether the animal chose the high-cost or the low-cost offer. In all cases, "cost-overt" offer types were those in which the animal chose the low-cost offer more frequently ( > 10%) than the high-cost offer, conditioned on the animal choosing each option at least twice; "cost-covert" offer types were those in which the animal consistently chose the same option independently of the action cost. To calculate the activity profile (i.e., the spike density function), trials were separately aligned at offer presentation, at target presentation and at juice delivery. For each alignment and each trial, spike times, expressed in 1 ms resolution, were convolved with a Gaussian kernel of 40 ms width. To normalize activity profiles, we first subtracted the mean activity in the pre-offer time window and then divided by the mean activity averaged across the other eight time windows.

All ROC analyses were performed on raw spike counts, without time averaging or baseline correction. To identify choice-related signals (Fig. 6c, d), we defined three time windows following offer presentation: post-offer 1 (0–250 ms), post-offer 2 (250 500 ms), and post-offer 3 (500 750 ms). For Fig. 6c, we first identified "cost-overt" offer types then we divided trials in two groups depending on the chosen juice (preferred or non-preferred). The two groups were compared with an ROC, from which we measured the AUC. This AUC is the probability with which an ideal observer would correctly infer the choice of the animal from the activity of one chosen juice cell and is thus equivalent to the measure of choice probability defined for perceptual decisions[57,58]. To obtain a single AUC for each neuron, we averaged the AUC across offer types[59]. For Fig. 6d, we divided trials in two groups depending on the chosen cost (high or low). We computed the AUC for each cost-overt offer type and we averaged across offer types.

**Dimensional integration in value signals**. We examined the integration of multiple determinants into a single value signals for *offer value (juice)* responses. To do so, we defined two variants of the variable *offer value (juice)*—one cost-affected and one cost-independent. We sought to assess which variant better fit neuronal responses. For each response, we considered the two $R^2$ and we computed the difference $\Delta R^2 = R^2_{\text{cost-affected}} - R^2_{\text{cost-independent}}$ and we examined the distribution for $\Delta R^2$ across the population. We did not want to bias the results in favor of either variant. Thus, for this analysis we identified neuronal responses encoding the *offer value (juice)* as follows. For each response and each value variable, we considered the two $R^2$ obtained from the two variants, and we assigned the maximum $R^2$ to the response. We then assigned each response to one of the selected variables accordingly. To further quantify the effects of action cost on this population, each response was fitted against the offer value using an ANCOVA (parallel model) and grouping data by the action cost. As any neuron could be tuned in multiple time windows, we conservatively focused only on the post-offer time window, when *offer value (juice)* responses were most prevalent.

We conducted similar analyses on responses encoding the *offer value (cost)*. Specifically, we examined whether firing rates varied as a function of the juice type. We defined two variants of the variable *offer value (cost)*—one commodity-affected and one commodity-independent. The variable selection analysis provided the same results for both variants. We identified neuronal responses encoding the *offer value (cost)* in an unbiased way based on the maximum of the two $R^2$. Focusing on the time window immediately following the offer, we computed the difference $\Delta R^2 = R^2_{\text{commodity-affected}} - R^2_{\text{commodity-independent}}$ and we examined the distribution for $\Delta R^2$ across the population. To further quantify the effects of juice type on this population, each response was regressed against the variable *offer value (cost)* using an ANCOVA (parallel model) and grouping data by the juice type. We conservatively focused only on the post-offer time window, when *offer value (cost)* responses were most prevalent.

Finally, we examined responses encoding the *chosen value*. We defined two variants of *chosen value*—one cost-affected and one cost-independent—and we verified that the variable selection analysis provided the same results. We then identified *chosen value* responses in an unbiased way based on the maximum $R^2$. For each response, computed the difference in $R^2$, and we examined the distribution for $\Delta R^2$ across the population in the post-offer time window. *Chosen value* responses were further analyzed with an ANCOVA (parallel model). For each response, data were grouped by the action cost and firing rates were regressed against the variable *chosen value*.

**Code availability**. The code used for data analysis is available upon reasonable request.

**Reporting summary**. Further information on experimental design is available in the Nature Research Reporting Summary linked to this article.

## Data availability
The data that support the findings of this study are available upon reasonable request.

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

## Acknowledgements

We thank Heide Schoknecht for help with animal training and Katherine Conen for insightful discussions. We also thank Sebastien Ballesta, Katherine Conen, Ahmad Jezzini, Masaru Kuwabara, Alessandro Livi, Weikang Shi, and Jue Xie for comments on the manuscript. This research was supported by the National Institutes of Health (grant number R01-DA032758 to CPS), the National Natural Science Foundation of China (grants 31571102 and 91632106 to XC), the Program of Introducing Talents of Discipline to Universities (Ministry of Education of China, Base B16018), the Joint Research Institute Seed Grants for Research Collaboration from the NYU-ECNU Institute of Brain and Cognitive Science at NYU Shanghai (to X.C.) and the Science and Technology Commission of Shanghai Municipality (grants 15JC1400104 and 16JC1400101 to XC).

## Author contributions

XC and CPS designed the study; XC collected and analyzed the data; XC and CPS wrote the manuscript.

## Additional information

**Competing interests:** The authors declare no competing interests.

