## [Peer Review File · Nature Communications]

Reviewers' comments:

Reviewer #1 (Remarks to the Author):

The study Good-based economic decisions under variable action costs, by Cai and Padoa-Schioppa, addresses a big question, namely, whether economic decisions with variable effort (action costs) are made by consideration of outcome value adjusted by action costs (outcome or good-based representations) or by consideration of the action plan biased by outcome value (action based representations). There is evidence in support of each idea.

To answer this question, the authors recorded from neurons in orbitofrontal cortex (OFC) of rhesus monkeys while the monkeys performed a decision-making task. The monkeys chose between two visual displays (the 'offers') that signaled different juices in variable amounts, with variable action costs. Action cost was manipulated by varying saccade amplitude. A key aspect of the task was that the monkeys could evaluate the offers (juice type, amount and action cost) early in the trial, but only later were the potential targets for action revealed. Thus, motor preparation for the choice response was separated in time from the evaluation of the offer.

The work builds on a productive line of investigation carried out by this group. The manuscript is well written and tightly argued. The findings are compelling and clearly illustrated. First, the authors verified that the 'effort' (action cost) manipulation was effective. They then moved forward with analyzing the activity of OFC neurons. As one would expect based on past work from this lab, the authors carried out a principled set of analyses using multiple linear regressions to determine exactly what factors (e.g., spatial location of offers, spatial location of saccade target, juice type, etc.) were being encoded by OFC neurons during the task. The main neurophysiological findings were: 1) the three types of encoding identified in OFC in earlier studies with fixed action costs were again found (offer value, chosen value, chosen juice); and 2) two new types of encoding were observed (offer value (cost) and chosen cost). Importantly, two types of neurons could encode choice: 'chosen juice' and 'chosen cost'. Both chosen juice cells and chosen cost cells encoded the 'decision' after the offer and well before the target presentation.

Additional analyses addressed whether OFC neurons encoded juice and action costs in an integrated manner. Neurons encoding offer value (juice) did not integrate action costs. Neurons encoding offer value (cost) encoded combined juice type, juice quantity, and cost. Finally, neurons encoding chosen value encoded juice type and quantity early in the trial, then progressed to encoding juice type, quantity and cost later in the trial when saccade targets had been revealed.

The authors conclude that OFC neurons represent identities and values of goods in two reference frames: good-based and cost-based. Because the encoding of choice outcome was apparent before target presentation, the authors also conclude that decisions (i.e., value comparisons) were made in goods space.

The study offers a novel set of findings. The results are important in showing that neurons in OFC can indeed encode information about different factors (here, juice type, juice amount and effort costs) that might be used to guide choice behavior. The task design is clever, and the authors have succeeded in their aim to separate offer evaluation from action planning. As the authors make explicit (p. 12, lines 470-474), this set of findings does not rule out the possibility that other brain regions provide a source of inputs to OFC for action costs. Nor do they rule out the possibility that, in other conditions, motor systems participate in value comparison. Although the results show that OFC neurons can encode choices early in the trial, this does not mean that the choice behavior is not dictated by a distributed network (e.g., 'multiple competitions taking place in parallel within and across brain regions'). Although the present results are impactful, and certainly help advance ideas regarding the goods- vs. action-based views of decision making, they would be even more impactful if the authors could discuss how these and other results help resolve the 'good-based view' vs. the 'distributed view'.

Specific comments: major:

- 1) The task combined the information about juice type and action cost into a single symbol (e.g., red plus sign). It is possible (indeed it seems likely) that this aspect of the design biased the OFC activity toward an integrated encoding of juice plus action costs. Do the authors have any insight into possible experimental outcomes had the effort costs been conveyed a different way \diamond (e.g., tactile or auditory cues, or even a separate set of visual cues)?
- 2) Although they have shown that OFC neurons can encode choices early in trial, this does not mean that the choice is not dictated by a distributed network (e.g., 'multiple competitions taking place in parallel within and across brain regions'). How would the authors suggest resolving the good-based view vs. the distributed view?
- 3) A recent paper reported that OFC neurons encoded the value of attended stimuli, independent of choice (Xie, Nie and Yang, eLife, 2018). The authors might want to add a caveat to their list of caveats (p. 12, lines 470-474) to the effect that, just because neurons encode chosen value etc. does not mean that those neurons (and their recorded activity) guide choice.
- 4) The idea that frontal cortex neurons/circuits 'can reconfigure itself depending on demands of the choice task' is not new. Earl Miller and others have noted that frontal cortex neurons encode several different factors, and the factors observed at any given time depend on task demands.

Specific comments: minor:

- 1) p. 12, line 466: edible instead of eatable?
- 2) p. 12, lines 486-488: what is the citation for the rat study?
- 3) p. 12, line 489-490: The citation in this sentence (ref 37) is for a study in which neurons were recorded in monkey amygdala, which was confusing given that the sentence topic is ACC. Is this as intended?
- 4) The discussion surrounding the Hayden study (Blanchard et al. ref 14) on p. 14 seems a bit long, and I was hard pressed to understand why this para is so important for the interpretation of the present findings. The authors should either explain why this information is relevant to their own findings or greatly reduce the length of this section.
- 5) p. 16, Methods: were the same juice types used every day? Or, alternatively, were different sets of juices used across the recording sessions? This information should be included.
- 6) It was not immediately obvious that the notation 'offer value -' (p. 7, line 240) translated to 'offer value (cost)'. And similarly that offer value B translated to 'offer value (juice)'. As a kindness to the reader this should be spelled out somewhere.

Reviewer #2 (Remarks to the Author):

This study looks at how action costs influence decision-making and neuronal responses in the orbitofrontal cortex (OFC). The behavioural design nicely separates the concept of action cost (effort) from action planning, and orthogonalizes cost with two other factors – reward (juice) taste and amount. Previous studies have considered some of these variables in choice tasks, but the current design allows a more in-depth analysis of how effort as a measurable quantity interacts with other sources of value information.

The main results are (1) that action costs are factored into the monkeys' decisions, (2) OFC neurons encode the costs independent of and before action plans, and (3) some neurons uniquely

encode costs, while others integrate cost with other sources of value information. The first two of these take-aways are reasonably sound, though not surprising given the previous literature. I have concerns about the methods leading to the third conclusion, although the finding that both integration and separation occur is fairly general and may not change. My overall impression is that the study is well conceived and designed, but the results are a predictable mixed bag of neural responses.

Regarding the section on integrating costs with other value information, the analysis seems cumbersome and the results are not very compelling one way or the other. To elaborate, in Fig 7 the shifts in the distributions are slight, so that some neurons have an index opposite to the reported population shift. This seems a shaky basis to draw sweeping conclusions like “neurons encoding the offer value (juice) did not integrate action costs with the other determinants of value”. In fact, these plots look like some neurons did integrate costs, but there was a small population-level bias in the other direction. Moreover, the effect sizes driving the population shift look very small. I understand that the authors don't feel confident that their variable selection approach can reliably quantify effect sizes in single neurons, but the task design has orthogonalized cost and other measures so it seems that it should be straight-forward to determine whether costs account for significant variance in neural firing. For example, maybe variance partitioning would be a reliable approach?

If the authors were able to obtain a more in-depth description of whether/how costs are integrated, it would be interesting to know more about the effects mentioned in the last panels of figure 7, where there seems to be a shift toward integrating cost later in the trial. Are these the same neurons that shift their coding schemes? Or are these a different population of neurons?

Another concern with focusing only on population analyses in figure 7 is that sometimes one neuron was counted more than once, since responses from multiple epochs are pooled. However, these aren't actually independent observations.

Minor

It should be clarified in the table or text that “chosen value” does not include cost. Also, why shouldn't “chosen value” include all measures that contribute to the choice? As it is now, shouldn't the chosen value variable be considered in the commodity reference frame (Table 2)?

In figure 2, there is a difference between the relationships depicted in the 3 panels, whereby neurons in B & C appear linearly related to the variable as stated, but the one in A is not very convincing. Granted this is just an example neuron, but is this a proper assignment of this neuron's activity? It's clearly not affected by cost, but in the first panel it appears only noisily active when A is chosen and quiescent when B is chosen.

Could the authors make the distinction between ‘overt’ and ‘covert’ clearer where it is introduced in the text? The labels aren't intuitive (maybe congruent/incongruent is better?) so I had to keep re-referencing what these meant. Also, it would help to clarify why the authors were concerned about this potentially changing the interpretation of the chosen juice signals.

Line 490 – I think the reference is incorrect

Reviewer #3 (Remarks to the Author):

Cai et al. examined neural mechanisms underlying decision-making when available goods are associated with variable values and actions costs. They specifically tested two supposedly competing hypotheses, 1) decisions are action-based and take place in premotor regions, or 2)

action costs are integrated with other determinants of value in an abstract representation and thus, decisions take place in goods space. Based on the observation that neurons in the orbitofrontal cortex (OFC) encode the chosen goods before the action target is instructed, they conclude that decisions are made in abstract, goods space.

The experiments and analyses clearly show that OFC represents the identity and values of chosen goods before the action targets are specified. However, I have fundamental issues regarding the setup of the hypotheses and the interpretation of the results as elaborated below. Even after the issues are resolved, in my opinion, the topic (i.e., the reference frame of decision) would better serve specialized groups in the field of decision making, rather than broader audience in Nature Communications.

My major concern is that it is not at all clear why the two hypotheses that the authors set up to test should be mutually exclusive. Even if decisions are made in action-based reference frame in premotor regions, isn't it possible that the decision information influences OFC such that it can represent the chosen goods? In other words, the representation of chosen goods in OFC itself does not seem to support or reject either hypothesis. Perhaps, the authors are thinking that decisions in their task could not have been action-based because they were made before the saccade target locations were specified. However, it has been shown that sensorimotor areas can prepare probable, multiple motor plans simultaneously when action targets are not specified as long as their potential locations are known (Cisek and Kalaska 2005; Klaes et al., 2011). The task in this paper is such a case. The subjects must have learned over the course of training that potential targets are 4 locations in the outer ring, and 4 locations in the inner ring. Thus, before the target presentation, premotor regions could start planning 8 potential motor plans. Then, each plan can integrate the value information derived from the offer type once the offer type is presented. As a result, action cost and values could be computed in action-based frame and the ones with the best value-cost outcome may survive. For example, saccade plans to 4 outer targets may sustain when offered values for higher cost are sufficiently large. This is equivalent to choosing the good associated with the higher cost although the specific target has not been finalized. Therefore, if OFC integrates this action-based decision made in premotor region and value information arisen within OFC, it can represent chosen goods even before the saccade targets are specified. Furthermore, integration with such feedback signal from the premotor region might explain why chosen juice representation in OFC appeared later when action costs were actually used for decision (overt trials) than not (Fig. 6A).

Indeed, the authors also acknowledged that varying the action costs introduce a significant challenge because the offer presentation had to instruct the subjects about the action cost while preventing the animal from planning the action itself. I agree with the authors, and for the same reason I cannot rule out the possibility that animals were planning multiple, potential saccades before the target was specified. Thus, although the results are clear and interesting, the interpretation thereof should be revised or better substantiated.

Specific comments:

Most of analyses were well explained and justified, and I have only a few questions/comments.

1. Please specify the cost-affected/commodity-affected variants in N-delta-R2 analysis. Are they the variables listed in rows 4-6 of Table 2?
2. What does it exactly mean to obtain the identical results in variable selection analysis between cost-affected and cost-independent variables in N-delta-R2 analysis? Does that mean for example, for a cell that the variable selection analysis selected the offer values of juice A, the variable selection analysis on cost-affected variants selects offer values of juice A if A is high-cost and/or offer values of juice A if A is low-cost?
3. Why is N-delta-R2 negative instead of zero when the additional information is not integrated in cell response?

4. The plots in the left column of Figure 2 and 3 should label the right side y axis with the probability to choose juice B.

Reviewer #1

The study *Good-based economic decisions under variable action costs*, by Cai and Padoa-Schioppa, addresses a big question, namely, whether economic decisions with variable effort (action costs) are made by consideration of outcome value adjusted by action costs (outcome or good-based representations) or by consideration of the action plan biased by outcome value (action based representations). There is evidence in support of each idea. To answer this question, the authors recorded from neurons in orbitofrontal cortex (OFC) of rhesus monkeys while the monkeys performed a decision-making task. The monkeys chose between two visual displays (the 'offers') that signaled different juices in variable amounts, with variable action costs. Action cost was manipulated by varying saccade amplitude. A key aspect of the task was that the monkeys could evaluate the offers (juice type, amount and action cost) early in the trial, but only later were the potential targets for action revealed. Thus, motor preparation for the choice response was separated in time from the evaluation of the offer. The work builds on a productive line of investigation carried out by this group. The manuscript is well written and tightly argued. The findings are compelling and clearly illustrated. First, the authors verified that the 'effort' (action cost) manipulation was effective. They then moved forward with analyzing the activity of OFC neurons. As one would expect based on past work from this lab, the authors carried out a principled set of analyses using multiple linear regressions to determine exactly what factors (e.g., spatial location of offers, spatial location of saccade target, juice type, etc.) were being encoded by OFC neurons during the task. The main neurophysiological findings were: 1) the three types of encoding identified in OFC in earlier studies with fixed action costs were again found (offer value, chosen value, chosen juice); and 2) two new types of encoding were observed (offer value (cost) and chosen cost). Importantly, two types of neurons could encode choice: 'chosen juice' and 'chosen cost'. Both chosen juice cells and chosen cost cells encoded the 'decision' after the offer and well before the target presentation. Additional analyses addressed whether OFC neurons encoded juice and action costs in an integrated manner. Neurons encoding offer value (juice) did not integrate action costs. Neurons encoding offer value (cost) encoded combined juice type, juice quantity, and cost. Finally, neurons encoding chosen value encoded juice type and quantity early in the trial, then progressed to encoding juice type, quantity and cost later in the trial when saccade targets had been revealed. The authors conclude that OFC neurons represent identities and values of goods in two reference frames: good-based and cost-based. Because the encoding of choice outcome was apparent before target presentation, the authors also conclude that decisions (i.e., value comparisons) were made in goods space.

The study offers a novel set of findings. The results are important in showing that neurons in OFC can indeed encode information about different factors (here, juice type, juice amount and effort costs) that might be used to guide choice behavior. The task design is clever, and the authors have succeeded in their aim to separate offer evaluation from action planning. As the authors make explicit (p. 12, lines 470-474), this set of findings does not rule out the possibility that other brain regions provide a source of inputs to OFC for action costs. Nor do they rule out the possibility that, in other conditions, motor systems participate in value comparison. Although the results show that OFC neurons can encode choices early in the trial, this does not mean that the choice behavior is not dictated by a distributed network (e.g., 'multiple competitions taking place in parallel within and across brain regions'). Although the present results are impactful, and certainly help advance ideas regarding the goods- vs. action-based views

of decision making, they would be even more impactful if the authors could discuss how these and other results help resolve the 'good-based view' vs. the 'distributed view'.

We thank Reviewer 1 for the constructive comments.

Specific comments: major:

1) The task combined the information about juice type and action cost into a single symbol (e.g., red plus sign). It is possible (indeed it seems likely) that this aspect of the design biased the OFC activity toward an integrated encoding of juice plus action costs. Do the authors have any insight into possible experimental outcomes had the effort costs been conveyed a different way \diamond (e.g., tactile or auditory cues, or even a separate set of visual cues)?

[Major 1] Interesting question. No, we don't have any particular insight. Our hunch is that the way we communicate with the animal does not matter, provided that the animal fully understands what the offers are. In other words, our hypothesis would be that the representation of value would be essentially the same if juice quantity and action cost were represented in different ways. But of course this is an open question for future studies.

2) Although they have shown that OFC neurons can encode choices early in trial, this does not mean that the choice is not dictated by a distributed network (e.g., 'multiple competitions taking place in parallel within and across brain regions'). How would the authors suggest resolving the good-based view vs. the distributed view?

[Major 2] Let us distinguish between how decisions are made in our study and how decision may be made in different situations. We argue that in our study the decision (i.e., the comparison between values) is good-based, because our task design prevented the animal from planning a movement before target presentation. This result is non-trivial, because most authors who wrote about this issue assumed that decisions under variable action cost would be action-based.

Conversely, as we clarify in the Discussion, "our results do not exclude that in different conditions – e.g., when offer presentation and action planning are not dissociated – motor systems may participate in value comparison". Testing whether this is the case – i.e., whether decisions are distributed when offer and action planning are not dissociated – is not easy. To be honest, we feel that the burden of designing and conducting such tests should be on those scholars who proposed the distributed decision model in the first place. In our understanding, the idea of distributed decisions is generally popular because it is ecumenical, and essentially vindicates every other proposal. At the same time, as far as we know, no one has ever provided direct evidence for a decision process taking place in multiple representations in parallel. Doing so requires showing that a particular decision cannot be explained as taking place exclusively in either the action-based or the good-based representation. Again, proving such case experimentally is not easy. As we suggested in a recent Review Article (Padoa-Schioppa and Conen, 2017), one possible approach is to use optogenetics to selectively excite or inhibit neurons associated to a particular action. If motor regions and the OFC are part of a distributed decision network, then manipulating the activity of neurons in motor regions

should predictably affect neuronal responses in OFC, and also predictably affect behavioral measures. Importantly, the manipulation would have to be sub-threshold.

3) A recent paper reported that OFC neurons encoded the value of attended stimuli, independent of choice (Xie, Nie and Yang, eLife, 2018). The authors might want to add a caveat to their list of caveats (p. 12, lines 470-474) to the effect that, just because neurons encode chosen value etc. does not mean that those neurons (and their recorded activity) guide choice.

[Major 3] We completely agree with the Reviewer – whether and how chosen value cells contribute to the decision remains unclear, and this is true independent of the results of Xie et al. The original ms already had a passage conveying this point. To further clarify this issue, we now rephrased it as follows (p.13):

In general, the role played by chosen value cells in the decision process remains unclear. Future work should further examine this important issue in the light of current observations.

4) The idea that frontal cortex neurons/circuits ‘can reconfigure itself depending on demands of the choice task’ is not new. Earl Miller and others have noted that frontal cortex neurons encode several different factors, and the factors observed at any given time depend on task demands.

[Major 4] On p.14 we added a sentence to acknowledge the fact that the neuronal malleability found here is consistent with established concepts about PFC, and we cited Miller and Cohen (2001).

Specific comments: minor:

1) p. 12, line 466: edible instead of eatable?

[Minor 1] We made the correction.

2) p. 12, lines 486-488: what is the citation for the rat study?

[Minor 2] We have now cited the correct study.

3) p. 12, line 489-490: The citation in this sentence (ref 37) is for a study in which neurons were recorded in monkey amygdala, which was confusing given that the sentence topic is ACC. Is this as intended?

[Minor 3] This was not intended! Thanks for catching the mistake, which we corrected.

4) The discussion surrounding the Hayden study (Blanchard et al. ref 14) on p. 14 seems a bit long, and I was hard pressed to understand why this para is so important for the interpretation of the present findings. The authors should either explain why this information is relevant to their own findings or greatly reduce the length of this section.

We significantly shortened that paragraph. However, we did not remove it completely because we think that it makes an important point. The Blanchard result is generally

perceived as a challenge to the idea that OFC represents integrated values, which of course is the theme of our current study. However, their main conclusion is problematic for the reasons we discuss here. Specifically, Blanchard et al drew their conclusion from an analysis that assumed an order-based reference frame. However, if the reference frame had been information-based (as their own data suggest it might have been), that analysis would not be relevant. Apart from setting the record straight on OFC, we want to highlight the importance of examining all the relevant reference frames.

As an aside, when the Blanchard paper came out in 2015, we wrote to the authors raising the possibility that the representation might be information-based. The authors never responded to our concerns. Subsequently, they published a secondary analysis of the same data, making the same claim, but still without addressing the possibility of different reference frames (Blanchard et al., 2018). Thus we believe that making our point here is important for the field.

5) p. 16, Methods: were the same juice types used every day? Or, alternatively, were different sets of juices used across the recording sessions? This information should be included.

[Minor 5] Different sets of juices were used. We added this information in the Methods (p.16).

6) It was not immediately obvious that the notation 'offer value –' (p. 7 , line 240) translated to 'offer value (cost)'. And similarly that offer value B translated to 'offer value (juice)'. As a kindness to the reader this should be spelled out somewhere.

[Minor 6] On p.6 we added a reference to Table 2, which defines all the variables included in the analysis.

Reviewer #2

This study looks at how action costs influence decision-making and neuronal responses in the orbitofrontal cortex (OFC). The behavioural design nicely separates the concept of action cost (effort) from action planning, and orthogonalizes cost with two other factors – reward (juice) taste and amount. Previous studies have considered some of these variables in choice tasks, but the current design allows a more in-depth analysis of how effort as a measurable quantity interacts with other sources of value information. The main results are (1) that action costs are factored into the monkeys' decisions, (2) OFC neurons encode the costs independent of and before action plans, and (3) some neurons uniquely encode costs, while others integrate cost with other sources of value information. The first two of these take-aways are reasonably sound, though not surprising given the previous literature. I have concerns about the methods leading to the third conclusion, although the finding that both integration and separation occur is fairly general and may not change. My overall impression is that the study is well conceived and designed, but the results are a predictable mixed bag of neural responses.

We thank Reviewer 2 for the constructive comments.

Regarding the section on integrating costs with other value information, the analysis seems cumbersome and the results are not very compelling one way or the other. To elaborate, in Fig 7 the shifts in the distributions are slight, so that some neurons have an index opposite to the reported population shift. This seems a shaky basis to draw sweeping conclusions like “neurons encoding the offer value (juice) did not integrate action costs with the other determinants of value”. In fact, these plots look like some neurons did integrate costs, but there was a small population-level bias in the other direction. Moreover, the effect sizes driving the population shift look very small. I understand that the authors don't feel confident that their variable selection approach can reliably quantify effect sizes in single neurons, but the task design has orthogonalized cost and other measures so it seems that it should be straight-forward to determine whether costs account for significant variance in neural firing. For example, maybe variance partitioning would be a reliable approach?

We addressed this question by performing the analysis of covariance (ANCOVA) and compared the effect of value and cost in Fig.S1. In essence, this analysis confirmed the results already reported in the original ms. The results of both analyses are described on p.10-11 (see here below).

If the authors were able to obtain a more in-depth description of whether/how costs are integrated, it would be interesting to know more about the effects mentioned in the last panels of figure 7, where there seems to be a shift toward integrating cost later in the trial. Are these the same neurons that shift their coding schemes? Or are these a different population of neurons?

We examined this issue with a chi-square test. We found that *chosen value* responses in post-offer and post-target time windows often came from different cells. This result is now reported on p.11 (see here below).

Another concern with focusing only on population analyses in figure 7 is that sometimes one neuron was counted more than once, since responses from multiple epochs are pooled. However, these aren't actually independent observations.

To address this issue, we repeated all the relevant analyses focusing only on one time window at a time.

The three concerns raised so far were addressed collectively by revising the relevant section of the Results (p.10-11), which now reads as follows:

The variable selection analysis described in the previous section was performed including the cost-affected variants of each value variable (see Table 2). However, we repeated the entire analysis using the cost-independent variants of each value variable, and we obtained identical results (i.e., the two procedures selected the same variables). To examine the effect of action cost on offer value coding and the relationship of these two factors on neural activity, neuronal responses encoding the *offer value (juice)* were fitted against offer value using an analysis of covariance (ANCOVA, parallel model) grouping data by the action cost. Since any neuron may be tuned in multiple time windows, we conservatively focused only on the post-offer time window, where *offer value (juice)* responses are most prevalent. The effect of cost was statistically significant ($p < 0.05$) for 15/60 (25%) responses. Across the population, there was no significant correlation between the slope of the encoding and the cost-related offset (Pearson correlation, $r = 0.12$, $p = 0.36$; Fig.S1A). To further investigate whether cost-affected or cost-independent variant provided a better fit of the neuronal responses, for each neuronal response encoding the *offer value (juice)*, we considered the two R^2 obtained for the two variants. We then computed their difference $\Delta R^2 = R^2_{\text{cost-affected}} - R^2_{\text{cost-independent}}$ and we examined the distribution for ΔR^2 across the population. We did not want to bias the results in favor of either variant. Thus for this analysis we identified neuronal responses encoding the *offer value (juice)* as follows. For each response and each value variable, we considered the two R^2 obtained from the two variants, and we assigned the maximum R^2 to the response. We then assigned each response to one of the selected variables accordingly. As illustrated in Fig.7A, the distribution of ΔR^2 was significantly displaced towards positive values (mean(ΔR^2) = -0.0065, $p = 0.055$, Wilcoxon signed-rank test). In other words, neurons encoding the *offer value (juice)* did not integrate action costs with the other determinants of value.

We conducted similar analyses on neurons encoding the *chosen value*. Their activity is known to vary as a function of both the juice type and the juice quantity (Padoa-Schioppa and Assad, 2006). Thus we examined whether their activity also varies as a function of the action cost. To do so, we defined two variants of the variable – one cost-affected and one cost-independent. We verified that the variable selection analysis provided the same results for both variants. We considered the R^2 obtained for the two variants of the variable. For this analysis, we identified neuronal responses encoding the *chosen value* in an unbiased way, based on the maximum of the two R^2 . Neuronal responses were submitted to an ANCOVA (parallel model). Focusing on the post-offer time window, the effect of cost was statistically significant ($p < 0.05$) for 12/45 (27%) responses. Across the population, there was no significant correlation between the slope of the encoding and the cost-related offset (Pearson correlation, $r = 0.21$, $p = 0.16$; Fig.S1B). Similarly, in the post-target window, the effect of cost was statistically significant for 7/25 (28%) responses. Across the population, there was no significant correlation between the slope of the encoding and the cost-related offset (Pearson correlation, $r = -0.27$, $p = 0.19$).

For responses classified as encoding the *chosen value*, we also computed the difference ΔR^2 and examined the distribution for ΔR^2 across the population. The distribution of ΔR^2 was significantly displaced towards negative values post-offer time window (mean(ΔR^2) = -0.013, $p < 0.01$, Wilcoxon signed-rank test; Fig. 7B) and towards positive values in post-target time window (mean(ΔR^2) = 0.0089, $p = 0.058$, Wilcoxon signed-rank test; Fig. 7B). The difference in ΔR^2 between early and late time windows was statistically significant (mean difference = 0.022, $p < 0.001$, Wilcoxon ranksum test; Fig. 7B). In other words, as a population, *chosen value* responses progressed from integrating only juice type and juice quantity before target presentation to integrating all three determinants of value (juice type, juice quantity and action cost) after the association between action and juice was revealed to the animal. However, *chosen value* responses in the two time windows often came from different cells (chi-square test, $p = 0.14$).

Lastly, we conducted a similar analysis on neurons encoding the *offer value (cost)*. In this case, each response is associated to a cost level, and the firing rate varies as a function of the juice quantity offered in any given trial. Thus, we examined whether the firing rate also varies as a function of the commodity (juice type). In essence, we repeated the analysis described above. We defined two variants of the variable – one commodity-affected and one commodity-independent. We verified that the variable selection analysis provided the same results for both variants of the variables. Focusing on the time window immediately following the offer, we considered the two R^2 obtained for the two variants of the variable (commodity-affected and commodity-independent). We identified neuronal responses encoding the *offer value (cost)* in an unbiased way, based on the maximum of the two R^2 . Thus for each response classified as encoding the *offer value (cost)* we computed the difference $\Delta R^2 = R^2_{\text{commodity-affected}} - R^2_{\text{commodity-independent}}$ and we examined the distribution for ΔR^2 across the population. In this case, the distribution of ΔR^2 was significantly displaced towards positive values (mean(ΔR^2) = 0.017, $p < 0.05$, Wilcoxon signed-rank test; Fig. 7C). In other words, these neuronal responses encoded the integrated variable that combined juice type and juice quantity in a cost-based reference frame.

Minor

It should be clarified in the table or text that “chosen value” does not include cost. Also, why shouldn’t “chosen value” include all measures that contribute to the choice? As it is now, shouldn’t the chosen value variable be considered in the commodity reference frame (Table 2)?

[Minor 1] Our null hypothesis is that value-related variables are computed incorporating action costs therefore all value-related variables listed in Table 2 include the cost term ξ computed from the animal’s choice behavior. The variable selected analysis reported in the main text was based on the variables defined in Table 2. However, as we discussed on p.10 “we repeated the entire analysis using the cost-independent variants of each value variable, and we obtained identical results.” Such results, on the other hand, indicate that the variable selection analysis does not serve the purpose of testing which variant fits the neural responses better. To this end, we performed model comparison by comparing the R^2 of the linear fit of cost-affected and cost-independent variant of the same variable (Figure 7).

In figure 2, there is a difference between the relationships depicted in the 3 panels, whereby neurons in B & C appear linearly related to the variable as stated, but the one in A is not very convincing. Granted this is just an example neuron, but is this a proper

assignment of this neuron's activity? It's clearly not affected by cost, but in the first panel it appears only noisily active when A is chosen and quiescent when B is chosen.

[Minor 2] We agree with Reviewer 2. We replaced Figure 2A with another cell encoding offer value B.

Could the authors make the distinction between 'overt' and 'covert' clearer where it is introduced in the text? The labels aren't intuitive (maybe congruent/incongruent is better?) so I had to keep re-referencing what these meant. Also, it would help to clarify why the authors were concerned about this potentially changing the interpretation of the chosen juice signals.

[Minor 3] We defined "cost-covert" and "cost-overt" offer types both in the Methods (p.18 under "**Activity profiles and ROC analysis.**") and in the section where relevant results were described (p.9). "In all cases, "cost-overt" offer types were those in which the animal chose the low-cost offer more frequently (>10%) than the high-cost offer, conditioned on the fact that the animal chose either option at least twice; "cost-covert" offer types were those in which the animal consistently chose the same option independently of the action cost."

We differentiated "cost-covert" and "cost-overt" offer types because in cost-covert conditions we don't have behavioral evidence that action costs were integrated into the decision process. To be conservative, we focused the analysis on "cost-overt" offer types, for which we have clear evidence from the animal's choice behavior that action cost was incorporated in the calculation of offer values.

Line 490 – I think the reference is incorrect

[Minor 4] We corrected the reference – thanks!

Reviewer #3

Cai et al. examined neural mechanisms underlying decision-making when available goods are associated with variable values and actions costs. They specifically tested two supposedly competing hypotheses, 1) decisions are action-based and take place in premotor regions, or 2) action costs are integrated with other determinants of value in an abstract representation and thus, decisions take place in goods space. Based on the observation that neurons in the orbitofrontal cortex (OFC) encode the chosen goods before the action target is instructed, they conclude that decisions are made in abstract, goods space. The experiments and analyses clearly show that OFC represents the identity and values of chosen goods before the action targets are specified. However, I have fundamental issues regarding the setup of the hypotheses and the interpretation of the results as elaborated below. Even after the issues are resolved, in my opinion, the topic (i.e., the reference frame of decision) would better serve specialized groups in the field of decision making, rather than broader audience in Nature Communications.

We thank Reviewer 3 for the constructive comments.

My major concern is that it is not at all clear why the two hypotheses that the authors set up to test should be mutually exclusive. Even if decisions are made in action-based reference frame in premotor regions, isn't it possible that the decision information influences OFC such that it can represent the chosen goods? In other words, the representation of chosen goods in OFC itself does not seem to support or reject either hypothesis. Perhaps, the authors are thinking that decisions in their task could not have been action-based because they were made before the saccade target locations were specified. However, it has been shown that sensorimotor areas can prepare probable, multiple motor plans simultaneously when action targets are not specified as long as their potential locations are known (Cisek and Kalaska 2005; Klaes et al., 2011). The task in this paper is such a case. The subjects must have learned over the course of training that potential targets are 4 locations in the outer ring, and 4 locations in the inner ring. Thus, before the target presentation, premotor regions could start planning 8 potential motor plans. Then, each plan can integrate the value information derived from the offer type once the offer type is presented. As a result, action cost and values could be computed in action-based frame and the ones with the best value-cost outcome may survive. For example, saccade plans to 4 outer targets may sustain when offered values for higher cost are sufficiently large. This is equivalent to choosing the good associated with the higher cost although the specific target has not been finalized. Therefore, if OFC integrates this action-based decision made in premotor region and value information arisen within OFC, it can represent chosen goods even before the saccade targets are specified. Furthermore, integration with such feedback signal from the premotor region might explain why chosen juice representation in OFC appeared later when action costs were actually used for decision (overt trials) than not (Fig. 6A).

We recognize the significance of this issue. Our response is in three parts.

(1) Earlier results referenced by Reviewer 3 are not as clear-cut as it might seem. In particular, the results of Cisek and Kalaska *do not* demonstrate that motor areas can prepare multiple motor plans at once. Such claim would be substantiated if the authors had found in motor regions neurons with bimodal tuning functions. The authors looked for such neurons but, strikingly, they did not find them. Indeed, in their task, cells in

motor and premotor cortices (areas F1 and F2) did not activate before the final instruction. Conversely, neurons that did activate prior to the final instruction were from prefrontal cortex (area F7) and thus most likely not motor (Picard and Strick, 2001). (For further discussion of Cisek and Kalaska's results, see Padoa-Schioppa (2011).)

(2) The evidence presented by Klaes et al. (2011) makes a stronger case for multiple motor plans. Still, it remains unclear whether neurons in PRR can be unequivocally considered as "motor", since neurons in parietal regions also have sensory/attentional responses. Also, the authors don't provide the exact locations of recordings in PMd, which opens the possibility that they too recorded from F7 as opposed to F2. Finally, in the free choice condition, animals had a strong and rather surprising bias in favor of the "inferred" movement, suggesting that the mental processes were more complex than schematized in the paper. Besides these caveats, the situation examined by Klaes et al was significantly simpler than that in our experiments. In their task, there were only two possible movements – one in the direction indicated by the preceding cue and one in the opposite direction. Furthermore, the animal did not need to use these possible movements as a reference frame to make a cognitive decision of any sort – the animal simply had to make one of the two movements. In contrast, in our experiments, there were always 8 possible movements, towards targets that had not been presented earlier in the trial. To make an action-based decision prior to target presentation, the animal should have computed 8 different action plans based on memory from earlier trials; associate 4 of these action plans (set A) with one offer value, associate the other 4 action plans (set B) with the other offer value; and then resolve the competition between two sets of action plans. The result of this competition would be a set of 4 action plans, which the animal would have to keep in working memory throughout the delay. Although it is theoretically conceivable, this scenario seems very unlikely, and we don't know of any empirical evidence suggesting that motor systems can support such complex cognitive operations. Hence, it seems much more reasonable to think that action-based decisions in our task would take place after target presentation.

(3) It is generally accepted in the literature that experimental manipulations of the sort we used here are a valid way to dissociate decision processes from action selection. Previous studies building on this concept focused on both perceptual decisions (Gold and Shadlen, 2003; Bennur and Gold, 2011) and economic decisions (Wunderlich et al., 2010; Cai and Padoa-Schioppa, 2014). Thus we believe that the logic of our study is sound and consistent with currently accepted notions. That said, we added to the Discussion a paragraph that discusses the issue raised by Reviewer 3 and summarizes the points made here (p.12).

Specific comments:

Most of analyses were well explained and justified, and I have only a few questions/comments.

1. Please specify the cost-affected/commodity-affected variants in N-delta-R2 analysis. Are they the variables listed in rows 4-6 of Table 2?

[1] All variables listed in Table 2 are cost-affected or commodity-affected, where applicable. In addition, we simplified the model comparison analysis by directly taking the difference of R^2 from the linear fitting of the two variants (cost-affected and cost-

independent) of the same variable as the index for model comparison. For example, the cost-affected variant of offer value B is $\#B + \xi \delta_{\text{juice B,+}}$ while the cost-independent variant is $\#B$.

2. What does it exactly mean to obtain the identical results in variable selection analysis between cost-affected and cost-independent variables in N-delta-R2 analysis? Does that mean for example, for a cell that the variable selection analysis selected the offer values of juice A, the variable selection analysis on cost-affected variants selects offer values of juice A if A is high-cost and/or offer values of juice A if A is low-cost?

[2] "Identical results" means the same sets of variables were selected. For variable selection analysis based on the cost-independent variants, we set ξ to 0 for all relevant variables, while for analysis based on cost-affected variants, ξ was computed from the animal's choice behavior using logistic regression (see Methods). The variable selection analysis selected the same sets of variables – offer value (juice), offer value (cost), chosen value, chosen juice and chosen cost.

3. Why is N-delta-R2 negative instead of zero when the additional information is not integrated in cell response?

[3] In the revised manuscript, we have simplified the model comparison analysis by directly taking the difference of R^2 from the linear fitting of the two variants (cost-affected and cost-independent) of the same variable as the index. For example, the cost-affected variant of offer value B is defined as $\#B + \xi \delta_{\text{juice B,+}}$ while the cost-independent variant is $\#B$. ΔR^2 being negative means the cost-independent variant provided a better fitting for the neural responses.

4. The plots in the left column of Figure 2 and 3 should label the right side y axis with the probability to choose juice B.

[4] We added the label.

References

- Bennur S, Gold JI (2011) Distinct representations of a perceptual decision and the associated oculomotor plan in the monkey lateral intraparietal area. *J Neurosci* 31:913-921.
- Blanchard TC, Piantadosi ST, Hayden BY (2018) Robust mixture modeling reveals category-free selectivity in reward region neuronal ensembles. *J Neurophysiol* 119:1305-1318.
- Cai X, Padoa-Schioppa C (2014) Contributions of orbitofrontal and lateral prefrontal cortices to economic choice and the good-to-action transformation. *Neuron* 81:1140-1151.
- Gold JI, Shadlen MN (2003) The influence of behavioral context on the representation of a perceptual decision in developing oculomotor commands. *J Neurosci* 23:632-651.
- Klaes C, Westendorff S, Chakrabarti S, Gail A (2011) Choosing goals, not rules: deciding among rule-based action plans. *Neuron* 70:536-548.
- Padoa-Schioppa C (2011) Neurobiology of economic choice: a good-based model. *Annu Rev Neurosci* 34:333-359.
- Padoa-Schioppa C, Assad JA (2006) Neurons in orbitofrontal cortex encode economic value. *Nature* 441:223-226.
- Padoa-Schioppa C, Conen KE (2017) Orbitofrontal cortex: A neural circuit for economic decisions. *Neuron* 96:736-754.
- Picard N, Strick PL (2001) Imaging the premotor areas. *Curr Opin Neurobiol* 11:663-672.
- Wunderlich K, Rangel A, O'Doherty JP (2010) Economic choices can be made using only stimulus values. *Proc Natl Acad Sci U S A* 107:15005-15010.

Reviewers' comments:

Reviewer #1 (Remarks to the Author):

The authors have adequately addressed my concerns.

After reviewing the authors responses to referees, together with the revised manuscript, I feel the authors have provided a measured and thoughtful responses to the queries from all three referees.

The study offers a novel set of findings showing that neurons in macaque OFC can encode values as an integrated quantity reflecting the three factors relevant to the decision: juice, juice quantity and action cost.

Reviewer #2 (Remarks to the Author):

I have read the revised manuscript. I think the revisions have generally improved the paper, and I only have a few additional comments that should be easy to address.

First, I think the new analyses presented in Fig. 7 are reasonable, but it was notable that $p = 0.055$ and $p = 0.058$ were both described as significant without qualification. In addition, 25-30% of individual neurons in each population significantly integrated costs. While I don't dispute the authors' overall conclusions, I think the manuscript needs to acknowledge that the results are not entirely clear-cut. For instance, borderline p-values are suggestive trends, and 25% of recorded neurons is not a negligible number.

Second, I was still momentarily confused when reading through the section about cost-overt and cost-covert trials. I think simply moving the lines defining cost-covert from 767-768 in the methods section to the paragraph in the results where the idea is introduced would greatly improve the reader's ability to quickly understand what the authors mean to convey.

Finally, I believe line 398 erroneously says the distribution was displaced toward positive values in Fig. 7A – this should read negative.

Reviewer #3 (Remarks to the Author):

I appreciate the authors for taking my comments seriously and offering their counter view in the response and incorporating it to the new discussion. Before giving specific comments to the counter view, I'd like to clarify my general position first. I am not fundamentally objecting the view that cost-based decision-making can take place in goods space. I also agree with the other reviewers that the paper presents valuable new data that will enrich the literature. But my issue is that the current data do not seem to refute the action-based decision hypothesis as the authors claim. So the paper should provide more rigorous interpretations of their results. The newly added discussion helps. But it still lacks scientific rigor and I think, such rigor should be used throughout the paper including the abstract, not just in the discussion.

The authors provide a three-part argument refuting my alternative interpretation of their results. Here are specific comments to their response in the corresponding order.

1) To refute the action-based decision hypothesis, the authors claimed that Cisek and Kalaska (2005) found neurons encoding potential action plans (bimodal tuning; PR cells) only in F7 but not F2, and thus there is no evidence for motor areas planning multiple action plans. I could not find this information in Cisek and Kalaska. Cisek and Kalaska certainly stated that PR cells were

distributed in gradient fashion such that they are more prevalent in the more rostral part of PMd and their most rostral part likely encroached the caudal part of F7. But nowhere in the paper could I find the assertion that PR cells were found only in F7 and none in F2. Less does not mean none, in my opinion. In fact, the paper indicated that the fraction of PR cells are 8% even in M1, which is more caudal to F2. I also read Padoa-Schioppa (2011) for further discussion, but did not find any information more than what was written in the authors' response. Perhaps the authors confirmed their assertion with Cisek and Kalaska in personal communication or some other objective way. If so, it should be stated in the discussion where they claimed that PR cells exist only in F7. Otherwise, I feel that the authors misinterpret Cisek and Kalaska.

2) The authors suspect that Klaes et al. (2011) might also have recorded cells encoding potential action plans from F7, instead of F2. Such speculation needs to be accompanied by substantiating evidence. It seems highly subjective and biased for the authors to argue that two published studies misidentified brain areas without presenting solid evidence.

3) In theory, one cannot really rule out the possibility that action costs are computed based on multiple, potential action plans if a finite number of action alternatives can be defined before the action specification. The authors seem to think that 8 potential plans are too many for the brain to handle. But as they conceded, in theory, it is possible. This does not mean that one cannot answer the question of where cost-based decision takes place. For instance, comparing the time course of information coding between OFC and motor areas in the same task may reveal that cost-based decisions indeed arise from OFC first or vice versa.

Taken together, I believe that the current data clearly show that cost-dependent decision is represented in goods space, which is a valuable new finding. However, the representation itself does not automatically indicate that it originates from good-based decision. The paper would be better received if the authors focus on describing the new findings in terms of representation, rather than fitting it for testing the two opposing hypotheses.

Reviewer #2

I have read the revised manuscript. I think the revisions have generally improved the paper, and I only have a few additional comments that should be easy to address.

We thank Reviewer 2 for the additional comments.

First, I think the new analyses presented in Fig. 7 are reasonable, but it was notable that $p = 0.055$ and $p = 0.058$ were both described as significant without qualification. In addition, 25-30% of individual neurons in each population significantly integrated costs. While I don't dispute the authors' overall conclusions, I think the manuscript needs to acknowledge that the results are not entirely clear-cut. For instance, borderline p-values are suggestive trends, and 25% of recorded neurons is not a negligible number.

We agree with Reviewer 2 and noted in the revised manuscript that the effects $p = 0.055$ and $p = 0.058$ are only tendential. For the percentage of responses with significant cost effect, we incorporated additional information with regard to whether the effect of cost is congruent or incongruent with the effect of offer value (juice) or chose value (juice), which provides insights for cost and value integration at the population level. We revised the relevant sections (p.10-11) as follows:

[...] The effect of cost was statistically significant ($p < 0.05$) for 15/60 (25%) responses, which can be congruent (8 responses) or incongruent (7 responses) with the effect of offer value (juice) (Fig.S1A).

[...] As illustrated in Fig.7A, the distribution of ΔR^2 was tendentially displaced towards negative values (mean(ΔR^2) = -0.0065, $p = 0.055$, Wilcoxon signed-rank test). In other words, neurons encoding the offer value (juice) have a tendency towards not integrating action costs with the other determinants of value.

[...] Focusing on the post-offer time window, the effect of cost was statistically significant for 12/45 (27%) responses, which can be congruent (2 responses) or incongruent (10 responses) with the effect of chosen value (juice) (Fig.S1B). Across the population, there was no significant correlation between the slope of the encoding and the cost-related offset (Pearson correlation, $r = 0.21$, $p = 0.16$; Fig.S1B). Similarly, in the post-target window, the effect of cost was statistically significant for 7/25 (28%) responses, which is mostly congruent (6/7 responses) with the effect of chosen value (juice) (Fig.S1C). Across the population, there was no significant correlation between the slope of the encoding and the cost-related offset (Pearson correlation, $r = -0.27$, $p = 0.19$; Fig.S1C).

[...] The distribution of ΔR^2 was significantly displaced towards negative values in the post-offer time window (mean(ΔR^2) = -0.013, $p < 0.01$, Wilcoxon signed-rank test; Fig.7B). In the post-target time window, the distribution of ΔR^2 was tendentially displaced towards positive values (mean(ΔR^2) = 0.0089, $p = 0.058$, Wilcoxon signed-rank test; Fig.7B).

Second, I was still momentarily confused when reading through the section about cost-overt and cost-covert trials. I think simply moving the lines defining cost-covert from 767-768 in the methods section to the paragraph in the results where the idea is introduced

would greatly improve the reader's ability to quickly understand what the authors mean to convey.

To improve clarity, we reiterated the definition of “cost-covert” and “cost-overt” trial types right before we introduced relevant analyses. We revised the relevant section (p.9) as follows:

Choices in our experiments reliably depended on the saccade amplitude, but the behavioral effect was relatively small (Fig.1C). Thus it is conceivable that upon easy decisions – i.e., decisions in which one of the juice values clearly dominated – monkeys effectively ignored the difference in action cost. One concern was whether the effect illustrated in Fig.6A was driven by trials in which animals ignored the action cost. **To address this issue, we defined "cost-overt" offer types as those in which the animal chose the low-cost offer more frequently (>10%) than the high-cost offer, conditioned on the fact that the animal chose either option at least twice. Conversely, "cost-covert" offer types were those in which the animal consistently chose the same option independently of the action cost.** We thus divided trials in four groups depending on the chosen juice (E or O) and on whether the effect of action costs was overt (o) or covert (c). For each group, we averaged the activity profiles across trials and across cells.

Finally, I believe line 398 erroneously says the distribution was displaced toward positive values in Fig. 7A – this should read negative.

We corrected it, thanks!

Reviewer #3

I appreciate the authors for taking my comments seriously and offering their counter view in the response and incorporating it to the new discussion. Before giving specific comments to the counter view, I'd like to clarify my general position first. I am not fundamentally objecting the view that cost-based decision-making can take place in goods space. I also agree with the other reviewers that the paper presents valuable new data that will enrich the literature. But my issue is that the current data do not seem to refute the action-based decision hypothesis as the authors claim. So the paper should provide more rigorous interpretations of their results. The newly added discussion helps. But it still lacks scientific rigor and I think, such rigor should be used throughout the paper including the abstract, not just in the discussion.

We thank Reviewer 3 for the additional comments. Before addressing them, we would like to make an important premise.

Following the previous revision of our manuscript, we became aware of a study recently published by Dekleva et al. (2018). The authors trained two monkeys to perform a task similar to that of Cisek and Kalaska (2005). They then recorded from PMd and M1 using chronically implanted arrays, and they conducted a series of analyses on the neuronal data. First, they conducted the same analysis of Cisek and Kalaska and replicated their results (bimodal responses). Second, they made a compelling case that these bimodal responses could be due to a statistical artifact. In essence, responses that look bimodal when averaged across trials might be unimodal on any given trial. Third, taking advantage of the fact that they had a large number of cells recorded simultaneously, they conducted a single-trial analysis. The results show that neurons in PMd process only one action plan at the time. Thus this new study calls into question the conclusions of Cisek and Kalaska and those of Klaes et al. In the light of the Dekleva study, it is fair to say that there is no compelling evidence that motor regions can represent multiple action plans at once. It is still worth discussing this hypothesis, but one should keep in mind that the hypothesis is based more on speculations than on empirical evidence.

The authors provide a three-part argument refuting my alternative interpretation of their results. Here are specific comments to their response in the corresponding order.

1) To refute the action-based decision hypothesis, the authors claimed that Cisek and Kalaska (2005) found neurons encoding potential action plans (bimodal tuning; PR cells) only in F7 but not F2, and thus there is no evidence for motor areas planning multiple action plans. I could not find this information in Cisek and Kalaska. Cisek and Kalaska certainly stated that PR cells were distributed in gradient fashion such that they are more prevalent in the more rostral part of PMd and their most rostral part likely encroached the caudal part of F7. But nowhere in the paper could I find the assertion that PR cells were found only in F7 and none in F2. Less does not mean none, in my opinion. In fact, the paper indicated that the fraction of PR cells are 8% even in M1, which is more caudal to F2. I also read Padoa-Schioppa (2011) for further discussion, but did not find any information more than what was written in the authors' response. Perhaps the authors confirmed their assertion with Cisek and Kalaska in personal communication or some other objective way. If so, it should be stated in the discussion where they claimed that PR cells exist only in F7. Otherwise, I feel that the authors misinterpret Cisek and Kalaska.

We recognize that the phrasing of our previous rebuttal was partially incorrect. However, the essence of our remarks was valid. Most of the bimodal cells of Cisek and Kalaska were recorded outside of and rostral to the arm region of PMd. Here below, we copied the anatomical reconstruction of Cisek and Kalaska (2005) (where bimodal cells are labeled "potential-response cells"), and the anatomical maps of Matelli and Luppino (2001) and Dum and Strick (2002). The two anatomical maps place the border between F7/PMdr and F2/PMdc in slightly different places – Dum and Strick's is aligned with the genu of the arcuate; Matelli and Luppino's is slightly more rostral. But in either case, it is clear that most of the bimodal neurons described by Cisek and Kalaska are from F7. Moreover, the recording locations of F2 bimodal neurons seem to be rather rostral, from a subregion at least partly overlapping with that denominated F2vr by Matelli and Luppino. In this subregion, electrical microstimulation elicits eye movements as opposed to arm movements (Fogassi et al., 1999; Raos et al., 2003). Since Cisek and Kalaska did not monitor the eye position, one cannot exclude that their bimodal cells reflect eye movements. Last but not least, the region where Cisek and Kalaska (2005) found bimodal cells seems to overlap with the region where Wallis and Miller (2003) found neurons encoding abstract (i.e., non-motor) rules. Hence, aside from other issues, bimodal cells could represent the task rule, as opposed to actual motor plans.

We revised the Discussion (p.12) as follows:

Previous studies suggested that motor regions can represent multiple action plans at once (Cisek and Kalaska, 2005; Klaes et al., 2011). Since we used a limited number of saccade target locations, one concern might be whether animals could make a decision early in the trial in an action-based representation. Several considerations argue against this hypothesis. First, evidence for the simultaneous representation of multiple action plans is not conclusive. One recurrent problem is that of trial averaging – responses that seem bimodal when averaged across trials might really be unimodal on any given trial (Dekleva et al., 2018). Indeed, one study that recorded from many cells simultaneously and conducted single-trial analyses concluded that neurons in premotor cortex process only one action plan at the time (Dekleva et al., 2018). Aside from this major issue, Cisek and Kalaska looked for but did not find bimodal neurons in F1; such neurons were found most prominently in rostral F2 and in F7 (Matelli and Luppino, 2001; Dum and Strick, 2002; Cisek and Kalaska, 2005). However, F7 is a prefrontal (not a motor) area (Picard and Strick, 2001), neurons in rostral F2 are often associated with eye movements rather than arm movements (Fogassi et al., 1999; Raos et al., 2003), and other studies found that neurons in this region can encode abstract rules as opposed to motor plans (Wallis and Miller, 2003). The study of Klaes et al (Klaes et al., 2011) seems to make a more compelling case, but recording locations were not specified and the problem of trial averaging remains. Moreover, the situation examined by Klaes and colleagues was significantly simpler than that in our experiments.

2) The authors suspect that Klaes et al. (2011) might also have recorded cells encoding potential action plans from F7, instead of F2. Such speculation needs to be accompanied by substantiating evidence. It seems highly subjective and biased for the authors to argue that two published studies misidentified brain areas without presenting solid evidence.

Unfortunately, Klaes et al did not provide an anatomical reconstruction of their recordings, so we don't know in what part of premotor area they were. Hence, we do not comment on this issue in the manuscript, except to note that recording locations were not specified.

3) In theory, one cannot really rule out the possibility that action costs are computed based on multiple, potential action plans if a finite number of action alternatives can be defined before the action specification. The authors seem to think that 8 potential plans are too many for the brain to handle. But as they conceded, in theory, it is possible. This does not mean that one cannot answer the question of where cost-based decision takes place. For instance, comparing the time course of information coding between OFC and motor areas in the same task may reveal that cost-based decisions indeed arise from OFC first or vice versa.

In the Discussion, we write that it is "theoretically conceivable" that monkeys perform our task by computing and comparing multiple action plans. In the light of the study of Dekleva et al. (2018), this assertion is, if anything, too generous. Indeed, as discussed above, there is really no convincing evidence that motor systems can ever represent more than one action plan at once, let alone representing many of them and using this representation to conduct value-based decisions. We agree with Reviewer 3 that comparing the time course of neural activity across brain areas would be useful, and we added a sentence to that effect in the Discussion (p.13). However, the key point is that this whole argument is about a highly speculative hypothesis.

Taken together, I believe that the current data clearly show that cost-dependent decision is represented in goods space, which is a valuable new finding. However, the representation itself does not automatically indicate that it originates from good-based decision. The paper would be better received if the authors focus on describing the new findings in terms of representation, rather than fitting it for testing the two opposing hypotheses.

Hopefully these responses will convince Reviewer 3 of the validity of our interpretation. In any case, we think that the paper presents the empirical results in a clear way, and that readers will be able to draw their own conclusions.

Cisek and Kalaska, 2005

Matelli and Luppino, 2001

Dum and Strick, 2002

Fig. 2. Motor areas in the frontal lobe of the macaque. Shaded regions indicate the location of the origin of corticospinal neurons projecting to the cervical segments of the spinal cord. Fine dotted lines indicate cytoarchitectonic borders of each premotor area and M1. Abbreviations:

References

- Cisek P, Kalaska JF (2005) Neural correlates of reaching decisions in dorsal premotor cortex: specification of multiple direction choices and final selection of action. *Neuron* 45:801-814.
- Dekleva BM, Kording KP, Miller LE (2018) Single reach plans in dorsal premotor cortex during a two-target task. *Nat Commun* 9:3556.
- Dum RP, Strick PL (2002) Motor areas in the frontal lobe of the primate. *Physiol Behav* 77:677-682.
- Fogassi L, Raos V, Franchi G, Gallese V, Luppino G, Matelli M (1999) Visual responses in the dorsal premotor area F2 of the macaque monkey. *Exp Brain Res* 128:194-199.
- Klaes C, Westendorff S, Chakrabarti S, Gail A (2011) Choosing goals, not rules: deciding among rule-based action plans. *Neuron* 70:536-548.
- Matelli M, Luppino G (2001) Parietofrontal circuits for action and space perception in the macaque monkey. *Neuroimage* 14:S27-32.
- Picard N, Strick PL (2001) Imaging the premotor areas. *Curr Opin Neurobiol* 11:663-672.
- Raos V, Franchi G, Gallese V, Fogassi L (2003) Somatotopic organization of the lateral part of area F2 (dorsal premotor cortex) of the macaque monkey. *J Neurophysiol* 89:1503-1518.
- Wallis JD, Miller EK (2003) From rule to response: neuronal processes in the premotor and prefrontal cortex. *J Neurophysiol* 90:1790-1806.

REVIEWERS' COMMENTS:

Reviewer #2 (Remarks to the Author):

I have read the authors' response and revised manuscript, and have no further concerns.

Reviewer #3 (Remarks to the Author):

I thank the authors for pointing me to the recent publication by Dekleva et al., which provides convincing evidence against simultaneous, multiple action plans in premotor regions. I agree that the new paper based on single-trial analysis compellingly raises the possibility that previous studies by Cisek and Kalaska or Klaes et al. might have mistook bimodal responses in their trial-averaged analysis as multiple action plans. In my opinion, the new findings by Dekleva et al. would sufficiently and concisely argue for the unlikelihood of multiple-plan-based cost computation in the premotor region, without speculating about the recording locations of the two previous studies in the discussion. Besides that, I have no further comments.